# Differentiable Conformal Training for LLM Reasoning Factuality

**Nathan Hittesdorf** [1]  **Marco Salzetta** [2]  **Lu Cheng** [1]

## Abstract

Large Language Models (LLMs) frequently hallucinate, limiting their reliability in critical applications. Conformal Prediction (CP) addresses this by calibrating error rates on held-out data to provide statistically valid confidence guarantees. Recent work extends CP to LLM factuality to filter out risky claims, ensuring that hallucination rates remain below a user-specified level (e.g., 10%). While prior methods treat claims independently, Coherent Factuality (Rubin-Toles et al., 2025) extends to multi-step reasoning by representing outputs as dependency graphs and jointly validating claims with their logical ancestors. A key limitation is that Coherent Factuality is not differentiable, requiring hand-crafted scorers that at high reliability levels remove nearly 60% of true claims. We introduce **Differentiable Coherent Factuality** (DCF), a fully differentiable relaxation that enables learning improved scorers while provably recovering the original algorithm's guarantees. Experiments on two benchmark reasoning datasets demonstrate DCF achieves up to **141%** improvement in claim retention while maintaining reliability guarantees, representing a significant step towards reliable conformal LLM systems.

## 1. Introduction

LLMs are increasingly deployed in high-stakes decision-making, making output factuality critical. However, LLMs are prone to hallucinations—confidently generating false information. Extensive work addresses this, including fine-grained atomic evaluation (Min et al., 2023), internal truthfulness detection (Azaria & Mitchell, 2023), and structured reasoning (Wei et al., 2022; Wang et al., 2023).

[1]Department of Computer Science, University of Illinois at Chicago, Chicago, United States [2]Department of Physics, University of Illinois at Urbana-Champaign, Urbana, United States. Correspondence to: Nathan Hittesdorf <nhitt2@uic.edu>, Lu Cheng <lucheng@uic.edu>.

*Proceedings of the 43$^{rd}$ International Conference on Machine Learning*, Seoul, South Korea. PMLR 306, 2026. Copyright 2026 by the author(s).

Recent approaches adapt Conformal Prediction (CP) (Vovk et al., 2005; Angelopoulos & Bates, 2023; Zhou et al., 2025)—a distribution-free method for constructing prediction sets with guaranteed error rates—to provide statistical factuality guarantees for LLMs (Mohri & Hashimoto, 2024; Cherian et al., 2024). These methods decompose outputs into atomic subclaims and assign each a *risk score* measuring likelihood of incorrectness. CP calibrates a threshold $\tau$ on held-out data, retaining only subclaims with scores below $\tau$, guaranteeing that retained outputs are factual with probability at least $1 - \alpha$ for user-specified error rate $\alpha$. Coherent Factuality (CF) (Rubin-Toles et al., 2025) extends this to multi-step reasoning via Approximate Deducibility Graphs (ADGs; Figure 6), where nodes are subclaims and edges encode logical dependencies. This captures contexts where claim validity depends on ancestor correctness, such as in mathematical reasoning.

A critical limitation of these approaches is the trade-off between reliability and retention: achieving stronger guarantees (smaller $\alpha$) requires more aggressive filtering. This is fundamental to CP—to guarantee fewer errors, the method must become more conservative. The problem is particularly acute with hand-crafted risk scores. While learned scoring functions can improve retention in standard CP settings (Cherian et al., 2024), CF relies on frequency-based self consistency scoring that cannot be optimized end-to-end. At high reliability levels ($\alpha < 0.1$), these hand-crafted approaches must remove up to 60% of true claims to maintain coverage guarantees, severely limiting practical utility (Figure 2a).

To address this limitation, we introduce **Differentiable Coherent Factuality (DCF)**, a fully differentiable relaxation of CF that enables gradient-based optimization of risk scores. Prior work has made conformal prediction differentiable for classification (Stutz et al., 2022) and independent claim filtering (Cherian et al., 2024), but these methods relax independent operations. CF's graph structure introduces tightly coupled discrete operations—threshold filtering, ancestor coherence enforcement, and argmax selection—that require joint relaxations preserving the algorithmic ordering (Section 3.1). We develop such relaxations and prove they recover the original CF algorithm in the limit (Theorems 3.1, 3.2), enabling training in a differentiable setting while preserving coverage guarantees at test time.

Our contributions are:

**(1)** We prove theoretically and demonstrate empirically that CF's discrete operations can be faithfully modeled via differentiable relaxations (Theorems 3.1, 3.2, Section 4.2).

**(2)** We show that the resulting framework enables end-to-end optimization of claim retention while preserving conformal coverage guarantees, achieving up to **141%** improvement on MATH and **61%** on FELM compared to frequency-based baselines (Sections 4.3).

**(3)** We provide in-depth interpretability analysis demonstrating that DCF learns to combine complementary signals more effectively than any individual feature (Section 4.4).

**Conflict of Interest Disclosure.** The authors declare no financial conflicts of interest.

## 2. Preliminaries

### 2.1. Conformal Prediction

Conformal Prediction (CP) (Vovk et al., 2005; Angelopoulos & Bates, 2023) provides distribution-free coverage guarantees under exchangeability. Given inputs $x \in \mathcal{X}$ and ground-truth labels $y \in \mathcal{Y}$, a *nonconformity score* $\nu_{\mathrm{cp}} : \mathcal{X} \times \mathcal{Y} \to \mathbb{R}$ measures how atypical a prediction is relative to the true label. In the classic split CP setting (Angelopoulos & Bates, 2023), we calibrate on a held-out set $\{(x_i, y_i)\}_{i=1}^n$ to compute threshold $\hat{\tau}_\alpha = \mathrm{Quantile}_{\lceil (1-\alpha)(n+1) \rceil / n}(\{\nu_{\mathrm{cp}}(x_i, y_i)\}_{i=1}^n)$. At test time, CP retains predictions with $\nu_{\mathrm{cp}}(x, y) \leq \hat{\tau}_\alpha$, guaranteeing $1 - \alpha \leq \mathbb{P}(y_{n+1} \in C(x_{n+1})) \leq 1 - \alpha + \frac{1}{n+1}$.

**Conformal factuality** (Mohri & Hashimoto, 2024) adapts CP to provide statistical guarantees on LLM outputs (illustrated in Figure 7). Given an LLM response, the method: (1) decomposes it into atomic subclaims—self-contained factual statements that can be independently verified, (2) assigns each claim a *risk score* $r_v$ measuring likelihood of incorrectness, and (3) retains only claims with $r_v \leq \hat{\tau}_\alpha$.

### 2.2. Coherent Factuality

Coherent Factuality (CF)[1] (Rubin-Toles et al., 2025) extend CP to multi-step reasoning. While Mohri & Hashimoto (2024) treats claims independently, CF recognizes that reasoning steps can only be evaluated within the context of preceding claims. For example, in proving $\sqrt{2}$ is irrational, the claim "$p^2$ is even" is only valid if the preceding claim "$p^2 = 2q^2$" is retained—CF enforces such dependencies.

**Approximate Deducibility Graphs.** CF represents outputs as ADGs $G = (V, E)$ where nodes are atomic subclaims

---

[1]We use CF to denote both the method and the property; context disambiguates.

and edges encode dependencies (Figure 6). A claim is *coherently factual* if it and all ancestors are correct.

**Subgraph Generation.** Given a risk score function $r : V \to \mathbb{R}$, CF generates candidate subgraphs $U_\tau$ for each threshold $\tau \in \mathcal{T}$ by: (1) selecting nodes with $r_v \leq \tau$, (2) removing nodes lacking ancestors, and (3) forming the induced subgraph.

**Nonconformity Score.** The nonconformity score is $\nu(X, Y, \mathcal{U}_\mathcal{T}) = \sup\{\tau : \text{all } U_{\tau'} \text{ with } \tau' \leq \tau \text{ are CF}\}$. The calibrated threshold $\hat{\tau}_\alpha$ is then obtained via split CP.

**Prediction.** At test time, the prediction set is $(U_{\mathrm{filtered}}, \tau_{\mathrm{filtered}}) = \arg\max_{(U, \tau) : \tau < \hat{\tau}_\alpha} \tau$.

**Scoring Function.** CF uses hand-crafted frequency-based scoring: $s(v) = (1 - \beta_{\mathrm{mix}}) f_{\mathrm{ind}}(v) + \beta_{\mathrm{mix}} \cdot \mathrm{med}\{f_{\mathrm{ind}}(v') : v' \in \mathrm{desc}(v)\}$, where $f_{\mathrm{ind}}(v)$ is self-consistency frequency and $\beta_{\mathrm{mix}} \in [0, 1]$ is a mixing parameter.

### 2.3. Conformal Training

Conformal Training (ConfTr) (Stutz et al., 2022) makes CP differentiable for classification by relaxing two operations: set construction (sigmoid-smoothed thresholding) and calibration (differentiable sorting for quantile computation). These relaxations are *independent*—each sample's confidence set is constructed in isolation. Cherian et al. (2024) extend this to LLM factuality with independent claim filtering. In both cases, at test time the original non-smooth CP is applied, preserving coverage guarantees.

## 3. Method

### 3.1. Overview and Motivation

CF's hand-crafted frequency-based scoring leads to poor retention at strict $\alpha$ levels—removing up to 60% of true claims. Conformal training offers a path forward by simulating the calibration and prediction pipeline during training to optimize scoring functions directly for claim retention. However, existing conformal training (Section 2.3) relaxes two independent operations—per-sample thresholding and quantile computation—where no prediction depends on another's outcome. CF's graph structure introduces three *coupled* discrete operations: **(1)** threshold filtering ($\mathbb{1}\{r_v \leq \tau\}$), **(2)** ancestor coherence (retaining a node only if all ancestors pass), and **(3)** argmax selection (a supremum over discrete sets shaped by prior steps). These form a cascade—each step's output depends on the previous step's discrete decisions—requiring joint relaxations that preserve this ordering.

**Our approach.** We develop DCF: differentiable relaxations that enable gradient flow while preserving CF's algorithmic structure. Our approach mirrors CF's calibration and prediction procedure, replacing each discrete operation

with a differentiable surrogate—sigmoids for indicators, products for conjunctions, and softmax for argmax. We then prove that these jointly recover the original CF algorithm in the limit (Theorems 3.1, 3.2), theoretically grounding the use of DCF for optimizing claim retention.[2]

**Proof challenges.** Proving that our relaxation recovers CF is non-trivial: the coupled temperature parameters controlling each relaxation must approach their limits in a specific order, since reversing the order corresponds to executing algorithmic steps out of sequence (e.g., taking a supremum before applying a sample cutoff). We address this by expressing all limits in terms of a single variable whose rate of approach encodes the correct ordering. An additional challenge arises when the supremum objective combines quantities on incomparable scales ($\tau$ values and violation indicators); we resolve this by restricting $\sup(\lambda\mathcal{T}) - \inf(\lambda\mathcal{T}) \leq 1$ to scale-match these terms. Full proofs appear in Appendix A.

### 3.2. Soft Membership

Calibration and prediction begin with the same two operations: threshold filtering and ancestor coherence (cf. **Subgraph Generation**). We relax these jointly into *soft membership* probabilities $q_{v,\tau} \in (0,1)$ representing the degree to which node $v$ belongs to the subgraph at threshold $\tau$.

**Soft Filtering.** CF selects nodes with $r_v \leq \tau$. For an ADG $\mathcal{G} = (V, E)$ and differentiable scorer $\pi_\theta : \mathbb{R}^d \to \mathbb{R}$, we compute confidence scores $\pi_\theta(\mathbf{x}_v)$ and risk values $r_v = C - \pi_\theta(\mathbf{x}_v)$, where $C$ is a constant.[3] Following Rubin-Toles et al. (2025), we construct threshold grid $\mathcal{T} = \{\tau_{\min}, r_{v_1}, \ldots, r_{v_n}, \tau_{\max}\}$ where $\tau_{\min} = \min_v r_v - m$ and $\tau_{\max} = \max_v r_v + m$.

The hard indicator for whether to retain a claim $v$ $\mathbb{1}\{r_v \leq \tau\}$ is non-differentiable. A natural relaxation is the sigmoid—a smooth approximation that outputs values in $(0,1)$ interpretable as retention probabilities $p_{v,\tau}$:

$$p_{v,\tau} = \sigma\left(\frac{\tau - r_v}{T_p}\right), \quad \sigma(x) = \frac{1}{1 + e^{-x}}. \quad (1)$$

Temperature $T_p$ controls sharpness: as $T_p \to 0^+$, the sigmoid recovers the hard indicator.

**Soft Ancestor Coherence.** CF removes nodes lacking ancestors—equivalently, a node is retained only if it *and all its ancestors* pass filtering. This conjunction is naturally relaxed as a product: $\Pr(\text{node } v) = \prod_{u \in \text{Anc}(v) \cup \{v\}} \Pr(u)$.

We implement this as a weighted geometric mean:[4]

$$\log q_{v,\tau} = \frac{\sum_{u \in \text{Anc}(v) \cup \{v\}} w_u \log p_{u,\tau}}{\sum_{u \in \text{Anc}(v) \cup \{v\}} w_u}, \quad (2)$$

where ancestors receive weight $w_u = \gamma$ and self receives $w_v = 1$. $\gamma > 0$ controls ancestor influence: $\gamma < 1$ attenuates, $\gamma = 1$ weights equally, $\gamma > 1$ amplifies. As a geometric mean of probabilities, $q_{v,\tau}$ approaches zero whenever any ancestor's retention probability approaches zero—preserving the conjunction.

### 3.3. Differentiable Calibration

Calibration computes a threshold $\hat{\tau}_\alpha$ guaranteeing $1 - \alpha$ coverage (cf. **Nonconformity Score**). For each calibration example, we find the maximum threshold at which the filtered graph contains no false claims—atomic subclaims that are hallucinated or factually incorrect. This defines the nonconformity score $\tilde{\tau}$. Taking a quantile across examples yields $\hat{\tau}_\alpha$.

With soft membership probabilities $q_{v,\tau}$ in hand, we must relax the discrete supremum $\sup\{\tau : \text{all } U_{\tau'} \text{ with } \tau' \leq \tau \text{ are CF}\}$. This requires two steps: measuring how well each threshold filters false claims, then selecting the largest threshold with acceptable filtering.

**Violation Measurement.** We quantify how well threshold $\tau$ filters false claims via a *violation score* $V_\tau \in [0,1]$, where higher values indicate the threshold is too permissive. Let $V^- = \{v : y_v = 0\}$ be the set of false claims. The validity score aggregates the probability of correctly filtering false claims:

$$\log Q_\tau = \frac{1}{|V^-|} \sum_{v \in V^-} \log(1 - q_{v,\tau}). \quad (3)$$

We convert validity to a violation measure:

$$V_\tau = 1 - \frac{Q_\tau^{1/\tau_s} - \text{softmin}\{Q_\tau^{1/\tau_s}\}}{\text{softmax}\{Q_\tau^{1/\tau_s}\} - \text{softmin}\{Q_\tau^{1/\tau_s}\} + \epsilon} \quad (4)$$

where $\tau_s > 0$ controls sharpness. As $\tau_s \to 0$ and $T_p \to 0$, this approaches hard violation detection.[5]

**Soft Supremum.** The CF nonconformity score is a supremum: the largest $\tau$ such that all subgraphs up to $\tau$ are CF. A supremum is an argmax, which we relax via softmax over a utility balancing threshold magnitude against violations:

$$s_\tau = \lambda \cdot \tau - V_\tau, \quad (5)$$

---

[2]Throughout Sections 3.2–3.4, we present formulas in log-space to match the implementation, where $\epsilon > 0$ is added to logarithm arguments for numerical stability.
[3]$C$ and margin $m$ are hyperparameters that aid comparison with CF.

[4]Geometric mean prevents chains from being punished for length; a standard product would dilute too aggressively.
[5]An alternative approach is to let $V_\tau = 1 - Q_\tau^{1/\tau_s}$. This also permits Theorem 3.1, but may produce different gradients.

where $\lambda > 0$ controls the trade-off. The soft supremum is:

$$w_\tau^{\text{cal}} = \frac{\exp(\beta \cdot s_\tau)}{\sum_{\tau' \in \mathcal{T}} \exp(\beta \cdot s_{\tau'})}, \quad \tilde{\tau} = \sum_{\tau \in \mathcal{T}} w_\tau^{\text{cal}} \cdot \tau. \quad (6)$$

As $\beta \to \infty$, the softmax concentrates on the maximum-utility threshold, recovering the hard supremum. Intuitively, the soft supremum finds the largest threshold that avoids retaining false claims—exactly the nonconformity score needed for calibration.

**Conformal Quantile.** The final calibration step computes $\hat{\tau}_\alpha$ as the $\lceil (1 - \alpha)(n + 1) \rceil / n$-quantile of the nonconformity scores $\{\tilde{\tau}_i\}_{i=1}^n$. In split CP, this is a simple order statistic; to enable gradient flow, we replace it with the differentiable soft quantile operator of Grover et al. (2019), which computes a smooth approximation $\hat{\tau}_\alpha = \text{SoftQuantile}(\{\tilde{\tau}_i\}, q, \rho)$ controlled by a sharpness parameter $\rho$. As $\rho \to \infty$, the soft quantile recovers the exact order statistic. This is the same mechanism used by ConfTr (Stutz et al., 2022) for differentiable calibration.

**Theorem 3.1** (Calibration Convergence). *As temperature parameters approach their limits ($T_p \to 0^+$, $\tau_s \to 0$, $\beta \to \infty$), the soft nonconformity score $\tilde{\tau}$ converges to the hard nonconformity score $\nu(\mathcal{U}_\mathcal{T})$ from CF. (Full statement and proof in Appendix A.1.)*

### 3.4. Differentiable Prediction

At test time, CF selects the maximum threshold below $\hat{\tau}_\alpha$ and returns the corresponding filtered subgraph (cf. **Prediction**): $(U_*, \tau_*) = \arg\max_{\tau < \hat{\tau}_\alpha} \tau$. We relax this constrained argmax while respecting the calibrated bound.

We first compute soft membership probabilities $q_{v,\tau}$ via Equations (1)–(2). Then we relax the constrained argmax using softmax combined with a sigmoid gate.

**Gated Soft Argmax.** The gate—another sigmoid relaxation of an indicator—smoothly transitions from 1 (below $\hat{\tau}_\alpha$) to 0 (above), enforcing the calibration bound:

$$w_\tau^{(\text{unnorm})} = \exp(\beta \cdot \tau) \cdot \sigma\left(\frac{\hat{\tau}_\alpha - \tau}{\tau_z}\right), \quad (7)$$

where $\exp(\beta \cdot \tau)$ weights toward larger thresholds and the sigmoid gates out thresholds exceeding $\hat{\tau}_\alpha$. Temperature $\tau_z$ controls gate sharpness: as $\tau_z \to 0^+$, the gate recovers the hard constraint $\tau < \hat{\tau}_\alpha$. After normalization, final retention probabilities $q_v$ aggregate across thresholds:

$$w_\tau = \frac{w_\tau^{(\text{unnorm})}}{\sum_{\tau' \in \mathcal{T}} w_{\tau'}^{(\text{unnorm})}}, \quad q_v = \sum_{\tau \in \mathcal{T}} w_\tau \cdot q_{v,\tau}. \quad (8)$$

**Theorem 3.2** (Prediction Convergence). *As temperature parameters approach their limits ($T_p \to 0^+$, $\tau_z \to 0^+$, $\beta \to \infty$), the soft retention probabilities $q_v$ converge to the hard CF prediction $U_{\text{filtered}}$. (Full statement and proof in Appendix A.2.)*

### 3.5. Training Objective

Following ConfTr (Stutz et al., 2022), we simulate the full conformal pipeline during training. At each epoch, we split the training data into disjoint calibration and prediction subsets $\mathcal{D}_{\text{cal}}$ and $\mathcal{D}_{\text{pred}}$. Differentiable calibration (Section 3.3) is applied to $\mathcal{D}_{\text{cal}}$ to obtain $\hat{\tau}_\alpha$, and differentiable prediction (Section 3.4) generates soft retention probabilities $q_{i,v}$ on $\mathcal{D}_{\text{pred}}$. Because all operations are differentiable, gradients flow from the loss through quantile estimation and calibration back to the scorer parameters $\theta$.

Following Cherian et al. (2024), we maximize the number of true claims retained:

$$\mathcal{L}_{\text{retention}} = -\frac{1}{|\mathcal{D}_{\text{pred}}|} \sum_{i \in \mathcal{D}_{\text{pred}}} \sum_{v \in V_i} y_{i,v} \cdot q_{i,v}, \quad (9)$$

where $y_{i,v} \in \{0, 1\}$ are ground-truth labels and $q_{i,v}$ are the coherence-adjusted soft retention probabilities from Equation (8). While the loss function is shared with Cherian et al. (2024), there it optimizes independent claim filtering; here, gradients flow through the full graph-structured pipeline—calibration, ancestor coherence, and gated prediction—enabling the scorer to learn patterns that exploit CF's dependency structure. This directly targets the limitation identified in Section 1: learning scores that retain more correct claims under conformal guarantees.

### 3.6. Summary and Training Procedure

Table 1 summarizes all transformations from CF to DCF. Theorems 3.1 and 3.2 establish that these relaxations jointly recover the original CF algorithm in the limit, ensuring that optimization of the differentiable surrogate corresponds to optimization of the discrete procedure. With convergence established, we formalize DCF into two differentiable subroutines—calibration (Algorithm 1) and prediction (Algorithm 2)—which compose into the end-to-end training procedure (Algorithm 3). At test time, the learned scorer deploys in the original discrete CF algorithm, preserving coverage guarantees. Full algorithm details are in Appendix B.

## 4. Experiments

As established in Section 1, hand-crafted scoring functions remove up to 60% of true claims at high reliability levels. Our experiments evaluate whether DCF's learned scoring can improve this retention-guarantee trade-off. We focus on $\alpha \in [0.01, 0.10]$—corresponding to 90–99% CF guarantees—where the trade-off is most acute and reliability matters most. Our experiments address three questions:

*Table 1.* Summary of transformations from CF (discrete) to DCF (differentiable).

| Operation | CF (Discrete) | DCF (Differentiable) | Limit |
|---|---|---|---|
| Threshold filtering | $\mathbb{1}\{r_v \leq \tau\}$ | $\sigma\left(\frac{\tau - r_v}{T_p}\right)$ | $T_p \to 0^+$ |
| Ancestor coherence | $v \in V_{\text{true}} \Rightarrow \text{Anc}(v) \subseteq V_{\text{true}}$ | $q_{v,\tau} = \text{GeomMean}(\{p_{u,\tau}\}_{u \in \text{Anc}(v) \cup \{v\}})$ | (conjunction) |
| Violation detection | $U_\tau$ is CF or not | $V_\tau = 1 - Q_\tau^{1/\tau_s}$ | $\tau_s \to 0$ |
| Nonconf. score | $\sup\{\tau : U_\tau \text{ is CF}\}$ | $\sum_\tau w_\tau \cdot \tau$ | $\beta \to \infty$ |
| Quantile | $\lceil (1-\alpha)(n+1) \rceil$-th order stat. | Soft quantile with strength $\rho$ | $\rho \to \infty$ |
| Prediction | $\arg\max_{\tau < \hat{\tau}_\alpha} \tau$ | $\sum_\tau w_\tau \cdot q_{v,\tau}, \quad w_\tau^{(\text{unnorm})} \propto \exp(\beta\tau) \cdot \sigma\left(\frac{\hat{\tau}_\alpha - \tau}{\tau_z}\right)$ | $\beta \to \infty, \tau_z \to 0$ |

- **RQ1 (Validity)**: Does DCF faithfully approximate CF to enable valid gradient-based optimization?
- **RQ2 (Performance)**: Does DCF retain more true claims than the SOTA methods while maintaining coverage guarantees?
- **RQ3 (Understanding)**: Does learning to *combine* features provide value beyond any single feature?

## 4.1. Experimental Setup

### 4.1.1. SCORER ARCHITECTURE

We implement $\pi_\theta$ as logistic regression over claim features to avoid overfitting and enable interpretability analysis.

### 4.1.2. DATASETS

We follow (Rubin-Toles et al., 2025) to use the following two benchmark datasets: (1) **MATH** (Hendrycks et al., 2021): 202 competition-level mathematics problems. We use GPT-5-mini (OpenAI, 2025) to generate solutions decomposed into atomic subclaims forming ADGs (Figure 6). Two annotators with strong math background performed dual-annotation to verify dependencies and label correctness. Graphs average 7.3 claims and 7.3 edges per problem, with 81.2% fully correct.(2) **FELM** (Chen et al., 2023): 710 problems across four domains: reasoning (207), math (194), world knowledge (184), and science (125), with subclaim-level decompositions and correctness annotations provided by the dataset authors. We exclude writing, which lacks structured logical dependencies. FELM exhibits simpler reasoning chains (4.0 claims, 2.8 edges per problem) with 62.4% fully correct.

**Features.** To provide training signals beyond frequency for the training of our scorer model, we extract 30 claim-level features organized into four categories (Cherian et al. (2024) similarly extract claim features for learned scoring): (1) base scoring (self-consistency frequency), (2) semantic coherence (logical consistency with ancestors, inference gap detection), (3) domain indicators (12 binary flags for mathematical topics), and (4) graph metrics (PageRank, betweenness, reachability computed via NetworkX). Semantic features use the same GPT-5-mini model as frequency scoring,

ruling out gains from stronger model access. For FELM, we exclude MATH-specific domain indicators and use two feature configurations: 7 features for $\alpha \leq 0.08$ and 20 features for $\alpha \geq 0.09$ (selected by correlation with ground-truth labels). See Appendix D.1 for complete feature descriptions.

### 4.1.3. EVALUATION PROTOCOL

We use 20-fold cross-validation to reduce variance in our estimates. Each fold partitions data into three subsets with distinct roles: (1) **Training set** (70%): Used to learn the scorer $\pi_\theta$, further split into internal calibration and prediction subsets for differentiable training (Figure 5 in Appendix C.1); (2) **Validation set** (15%): Serves a dual role—early stopping during training, and held-out calibration of the frozen scorer at test time. Scorer parameters are never updated using gradients from this set, preserving conformal exchangeability; (3) **Test set** (15%): Held-out for final evaluation; never used during training or calibration. At test time, the learned scorer (with frozen parameters) deploys in the original CF algorithm using the validation set as the calibration set, preserving the conformal coverage guarantee.

### 4.1.4. BASELINES

We compare DCF against state-of-the-art conformal LLM factuality methods, each ablating a key component of DCF:

- **Coherent Factuality (CF)** (Rubin-Toles et al., 2025): Hand-crafted frequency-based scoring with graph structure. *Does learning improve over hand-crafted scoring?*
- **Independent Factuality** (Mohri & Hashimoto, 2024): Frequency-based scoring without graph structure—claims treated independently. *Does modeling dependencies help?*
- **Boosted Independent** (Cherian et al., 2024): Learned scoring with the same features as DCF, but without graph structure. *Does learning alone suffice, or is graph structure necessary?*
- **XGBoost + Conformal**: XGBoost trained independently for optimizing CF accuracy on the same features, then plugged into CF as the scorer. *Can a standard classifier replace DCF's differentiable training?*

### 4.1.5. METRICS

Theorems 3.1 and 3.2 establish that DCF recovers CF in the limit. To verify this empirically—confirming that soft operations approximate hard operations well enough to train $\pi_\theta$ (RQ1)—we report **correlation metrics** (Pearson $r$, MAE) between soft and hard quantities. For evaluating performance (RQ2, RQ3), we follow standard CP evaluation (Angelopoulos & Bates, 2023) and LLM factuality evaluation (Rubin-Toles et al., 2025): (1) **Coverage**: Proportion of test examples where retained claims are CF; target is $1 - \alpha$. (2) **Retention**: Mean claims retained per problem (or equivalently, proportion of claims retained), measuring informativeness of filtered outputs.

### 4.2. RQ1: Validating Surrogate Behavior

We empirically verify Theorems 3.1 and 3.2 on MATH, focusing on problems with at least one incorrect claim where calibration is non-trivial (error-free problems yield constant thresholds uninformative for validation). We use MATH because (1) we performed manual annotation with direct control over label quality, and (2) its denser graphs (avg. 7.3 claims, 7.3 edges vs. FELM's 4.0 claims, 2.8 edges) make ancestor constraints more meaningful.

### 4.2.1. CALIBRATION VALIDATION

Training uses finite temperatures (Appendix D.3), so we validate empirically that these settings yield behavior sufficiently close to CF. Figure 1a shows strong correlation between soft and hard nonconformity scores ($r = 0.921$, MAE= 0.34). Figure 1b demonstrates near-perfect threshold agreement across $\alpha \in [0.01, 0.15]$ ($r = 0.976$, MAE= 0.12, 3% relative error), confirming the soft quantile preserves calibration properties.[6]

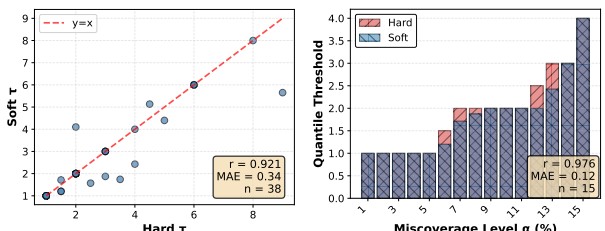

*(a)* Individual nonconf. scores.   *(b)* Agg. thresholds across $\alpha$.

*Figure 1.* Soft vs. hard calibration validation.

### 4.2.2. PREDICTION VALIDATION

Table 2 shows claim-level agreement across 20-fold CV (14,600 predictions). Agreement ranges from 90.2–100%,

with perfect agreement at $\alpha \leq 0.03$ and minor discrepancies at threshold boundary regions. This validates Theorem 3.2.

*Table 2.* Soft-hard prediction agreement across $\alpha$ values.

| $\alpha$ | Both Incl. | Soft Only | Hard Only | Both Excl. | Agree |
|---|---|---|---|---|---|
| 0.01–0.03 | 4791 | 0 | 0 | 9809 | 100.0% |
| 0.04 | 4791 | 0 | 30 | 9779 | 99.8% |
| 0.05 | 4813 | 0 | 902 | 8885 | 93.8% |
| 0.06 | 5428 | 2 | 888 | 8282 | 93.9% |
| 0.07 | 5848 | 4 | 668 | 8080 | 95.4% |
| 0.08 | 6206 | 8 | 1232 | 7154 | 91.5% |
| 0.09 | 6923 | 12 | 1416 | 6249 | 90.2% |
| 0.10 | 7455 | 54 | 994 | 6097 | 92.8% |

**Summary.** Both calibration ($r > 0.92$) and prediction ($> 90\%$ agreement) validations confirm that DCF with practical hyperparameters faithfully approximates CF, ensuring training gradients reflect the true discrete objective.

### 4.3. RQ2: Performance Comparisons

Having validated DCF as a faithful surrogate (RQ1), we evaluate whether the differentiable framework learns improved scoring functions. We compare DCF against the baselines introduced in Section 4.1, isolating whether improvements stem from learning, from modeling dependencies, or from their combination.

**Main Results.** Figure 2 presents retention and coverage across $\alpha \in [0.01, 0.10]$ on MATH and FELM respectively.[7] On MATH, DCF meets the coverage target for $\alpha \in 0.05, 0.06, 0.07, 0.08$, achieving 60–105% retention improvements over CF. At $\alpha = 0.03$ (97% reliability), DCF more than doubles retention (+141%, 1.76 vs. 0.73 claims) with coverage within 0.5pp of target. On FELM, DCF meets the coverage target at 9 of 10 $\alpha$ values and achieves higher retention than CF at 6 of those 9, with gains concentrated at lower $\alpha$ (0.01–0.05) and higher $\alpha$ (0.09–0.10). At mid-range $\alpha$ (0.06–0.08), CF achieves higher retention; however, DCF's coverage at these levels exceeds the target by 0.9–1.6pp (e.g., 93.64% vs. 92% target at $\alpha = 0.08$), suggesting room for trading excess coverage for improved retention. Complete numerical results appear in Tables 8 and 9 (Appendix D).

**Baseline Ordering Validates Design.** DCF consistently outperforms all baselines, including the three state-of-the-art conformal factuality methods. The ordering (DCF > CF > Independent > Boosted Independent $\gg$ XGBoost) confirms that learning improves over hand-crafted scoring (DCF vs.

---

[6]Since 18.8% of MATH problems contain errors, thresholds saturate at $\tau_{\max}$ for $\alpha \gtrsim 0.19$. We evaluate at $\alpha \leq 0.15$, below saturation.

[7]At $\alpha \in \{0.01, 0.02\}$ on MATH, only 18.8% of problems contain errors, leaving too few calibration examples to estimate quantiles at that granularity; our implementation falls back to $\tau_{\min}$ (retain none) to preserve the marginal guarantee. We report these values but focus analysis on $\alpha \geq 0.03$.

CF), that modeling dependencies adds value even without learning (CF vs. Independent), and that learning without graph structure actually hurts—Boosted Independent underperforms non-learned Independent while using identical frequency-score. XGBoost trained independently for claim-level accuracy achieves near-zero retention at tight coverage levels, showing that a standard classifier cannot replace DCF's differentiable conformal training. Together, these results demonstrate that DCF's gains require *both* learning and graph structure—neither component alone is sufficient—and that learning must occur *through* the conformal objective.

**Summary.** DCF consistently outperforms SOTA across both datasets and reliability levels by directly optimizing CF's conformal objective with learned, graph-aware scoring.

### 4.3.1. WHY DCF GAINS DIFFER ACROSS DATASETS

The retention improvements in Figure 2 are substantially larger on MATH than on FELM. We attribute this to a fundamental difference in how feature signal is distributed across the two datasets. We computed feature-wise mutual information (MI) with ground-truth claim correctness for both the frequency score and the orthogonal residual of all remaining 29 features (Table 3). On MATH, structural and graph features carry 0.089 nats of information beyond frequency—a 636% uplift over frequency alone. On FELM, the same residual features carry essentially zero orthogonal signal, while frequency alone is already an order of magnitude more informative (0.170 nats). DCF learns to exploit orthogonal signal that frequency misses; on FELM where this signal is largely absent, DCF's relative gains are correspondingly smaller (hypothesized mechanism for FELM's residual gains in Appendix E.5).

*Table 3.* Feature–claim-correctness mutual information (nats): frequency-score vs. orthogonal residual of the other 29 features. Orthogonal MI is computed as $\sum_f \max(0, I(f; y) - I(f; \text{freq}))$ via a kNN-based estimator (Kraskov et al., 2004); values are relative magnitudes, and "0" indicates orthogonal contribution below the estimator's resolution rather than a strict zero.

| Dataset | Frequency MI | Orthogonal MI | Uplift |
|---|---|---|---|
| MATH | 0.014 | **0.089** | +636% |
| FELM | 0.170 | 0.000 | +0% |

### 4.4. RQ3: Understanding DCF

We now ask whether DCF's gains stem from learning to *combine* features, or simply from accessing more discriminative individual features. CF can use any scalar feature as its risk score—the original method uses frequency, but graph metrics or positional features could substitute. We test whether any single feature, optimally configured, can match DCF.

### 4.4.1. SINGLE-FEATURE ABLATION

Using SHAP analysis (Appendix F.4), we identify DCF's most influential features: `nx_reachability` (graph connectivity), `claim_index` (position in reasoning chain), and `frequency-score`. We construct single-feature CF baselines using each as the risk score, with $\beta_{\text{mix}}$ optimized per-$\alpha$ via grid search.

Figure 3 shows DCF consistently outperforms single-feature baselines on both datasets. On MATH, DCF outperforms the best single feature (`nx_reachability`) by 7–50% across $\alpha$ values. On FELM, DCF outperforms frequency-score at most $\alpha$ values (by up to 64%); at mid-range $\alpha$ where frequency wins, DCF maintains excess coverage (Section 4.3). Notably, the best single feature differs by dataset—graph connectivity dominates on MATH's complex reasoning chains, while frequency suffices more on FELM's simpler structures—yet DCF matches or exceeds both.

### 4.4.2. CASE STUDY

To illustrate how combining features improves retention, consider a MATH problem at $\alpha = 0.06$ where all 12 claims are correct. The CF baseline (frequency scoring with optimized $\beta_{\text{mix}}$) retains 0/12 claims; DCF retains 9/12.

This problem's solution relies on Vieta's formulas—a valid but less common technique that the LLM rarely regenerates. As a result, key steps receive frequency $= 0$, causing the baseline to reject the entire chain. DCF succeeds by combining complementary signals: when one feature is unreliable, others compensate.

*Table 4.* DCF feature contributions for claim 8 (frequency=0).

| Feature | Value | Weight | Contrib. |
|---|---|---|---|
| `nx_reachability` | 3.00 | 0.272 | +0.815 |
| `claim_index` | 8.00 | 0.098 | +0.780 |
| `nx_in_degree` | 2.00 | 0.135 | +0.269 |
| `quadratic_equations` | 1.00 | 0.229 | +0.229 |
| `nx_out_degree` | 1.00 | 0.176 | +0.176 |
| `problem_relevance` | 1.00 | 0.144 | +0.144 |
| `coherent_to_ancestors` | 1.00 | 0.110 | +0.110 |
| `nx_betweenness` | 0.22 | 0.292 | +0.064 |
| **Total** | | | **2.66** |

Table 4 shows how DCF retains Claim 8 despite frequency $= 0$. Graph structure (`nx_reachability`, `claim_index`) provides strong positive signal, while connectivity and semantic features contribute additional evidence. The learned combination accumulates evidence across signals, enabling retention even when key features like frequency are unreliable. This flexibility is DCF's core advantage: rather than relying on one potentially unreliable signal, it learns which features to trust and how to weight them. See Appendix F for the full methodology and an additional case study.

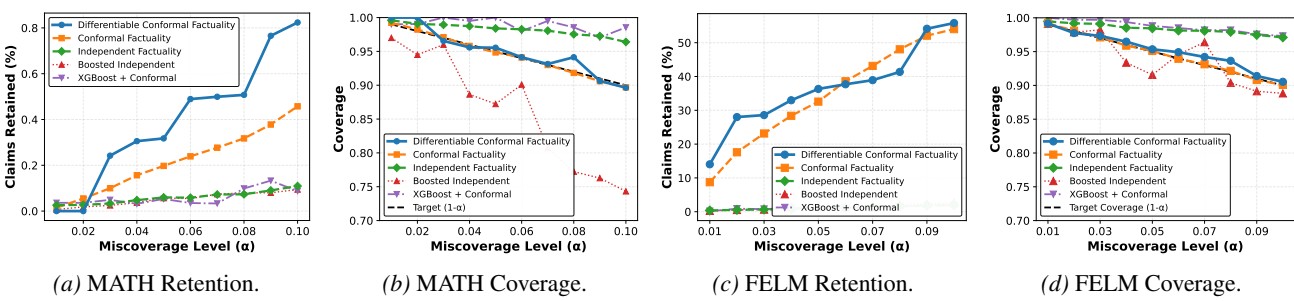

*Figure 2.* DCF vs. baselines on MATH (a–b) and FELM (c–d) across $\alpha \in [0.01, 0.10]$.

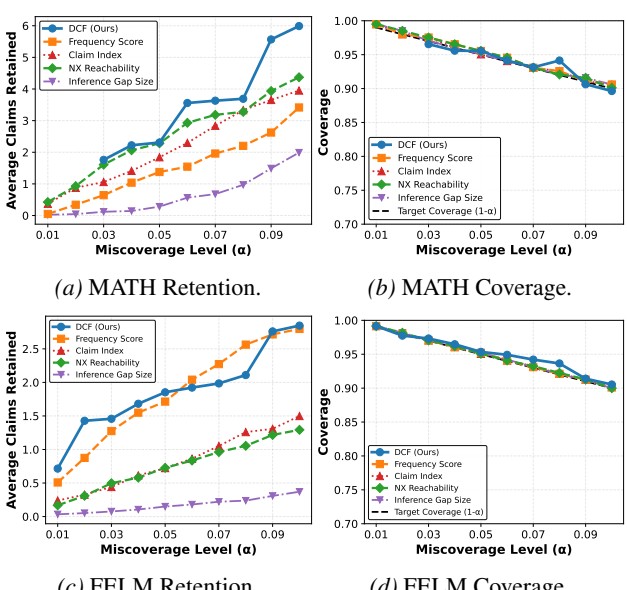

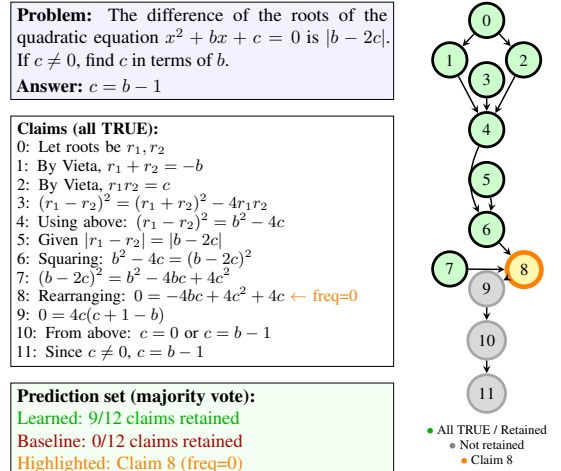

*Figure 3.* DCF vs. single-feature CF baselines (optimized $\beta_{\text{mix}}$) on MATH (top) and FELM (bottom).

**Problem:** The difference of the roots of the quadratic equation $x^2 + bx + c = 0$ is $|b - 2c|$. If $c \neq 0$, find $c$ in terms of $b$.
**Answer:** $c = b - 1$

**Claims (all TRUE):**
0: Let roots be $r_1, r_2$
1: By Vieta, $r_1 + r_2 = -b$
2: By Vieta, $r_1 r_2 = c$
3: $(r_1 - r_2)^2 = (r_1 + r_2)^2 - 4r_1 r_2$
4: Using above: $(r_1 - r_2)^2 = b^2 - 4c$
5: Given $|r_1 - r_2| = |b - 2c|$
6: Squaring: $b^2 - 4c = (b - 2c)^2$
7: $(b - 2c)^2 = b^2 - 4bc + 4c^2$
8: Rearranging: $0 = -4bc + 4c^2 + 4c$ ← freq=0
9: $0 = 4c(c + 1 - b)$
10: From above: $c = 0$ or $c = b - 1$
11: Since $c \neq 0$, $c = b - 1$

**Prediction set (majority vote):**
Learned: 9/12 claims retained
Baseline: 0/12 claims retained
Highlighted: Claim 8 (freq=0)

• All TRUE / Retained
• Not retained
• Claim 8

*Figure 4.* CF vs. DCF ADG Comparison (Correct Claims)

## 5. Related Work

**Conformal Prediction.** Conformal prediction (CP) (Vovk et al., 2005; Shafer & Vovk, 2008; Angelopoulos & Bates, 2023) provides distribution-free coverage guarantees under exchangeability. Extensions include conformalized quantile regression (Romano et al., 2019) and the learn-then-test framework (Angelopoulos et al., 2025), which calibrates predictive algorithms to achieve risk control—inspiring our treatment of conformal calibration as an optimization problem.

Most relevant to our work, Stutz et al. (2022) introduced ConfTr (Conformal Training), demonstrating that discrete quantile operations in CP can be relaxed into differentiable surrogates for end-to-end optimization—a key insight we extend to graph-structured reasoning. Angelopoulos et al. (2024) generalized this to conformal risk control with arbitrary monotone loss functions. Gibbs & Candes (2021) proposed adaptive conformal inference for distribution shift, though extending such adaptation to graph-structured dependencies remains open (Zhao et al., 2024).

**LLM Factuality.** Recent work has extended conformal methods to LLMs for statistically rigorous uncertainty quantification (Zhou et al., 2025). Kumar et al. (2023) and Quach et al. (2024) were among the first to adapt CP for LLM settings, while Su et al. (2024) introduced sampling-based nonconformity scores for API-only access and Wang et al. (2024) incorporated self-consistency into CP for correctness coverage. These methods operate at the response or question level; a parallel line of work applies CP to structured, claim-level factuality.

Mohri & Hashimoto (2024) pioneered conformal factuality for atomic claims but ignored logical dependencies. Rubin-Toles et al. (2025) extended this to coherent factuality with approximate deducibility graphs, but still relied on hand-crafted scoring that can be overly conservative.

Cherian et al. (2024) developed enhanced conformal methods with conditional validity and differentiable scoring functions, demonstrating retention improvements but operating on independent claims without exploiting graph

structure. Min et al. (2023) introduced FActScore for fine-grained atomic evaluation, directly informing our claim-level analysis.

## 6. Conclusion

We addressed the challenge of LLM factuality verification: existing conformal methods provide formal guarantees but sacrifice too many true claims to be useful. We introduced Differentiable Coherent Factuality (DCF), the first differentiable relaxation of graph-structured LLM conformal reasoning, which enables end-to-end optimization through calibration and prediction, provably recovers the original CF algorithm in the limit (with 93–100% empirical agreement), and achieves up to 141% retention improvement over frequency-based baselines while deploying directly in the standard CF algorithm at test time.

More broadly, DCF bridges formal guarantees and usability: statistical guarantees provide little value if outputs are too conservative to be informative. Once trained, a DCF scorer deploys on unlabeled IID data via the standard CF algorithm, making it a natural fit for monitoring agentic workflows—where dependency-graph structure is the dominant deployment pattern and where rejected claims can drive downstream repair (Section F.4).

## Acknowledgments

This work is supported by the National Science Foundation (NSF) Grant #2312862, NSF-Simons SkAI Institute, NSF CAREER #2440542, NSF #2533996, National Institutes of Health (NIH) #R01AG091762, NSF ACCESS Computing Resources, NAIRR, NRP, a Google Research Scholar Award, and a Cisco gift grant.

## Impact Statement

This work improves factual reliability of LLM outputs through formal statistical guarantees, contributing to safer AI deployment in accuracy-critical domains. We note that conformal guarantees are statistical (holding on average, not per-prediction). Additionally, as noted by Cherian et al. (2024), optimizing for marginal coverage may yield uneven reliability across subgroups or covariates, potentially misrepresenting performance on certain data subsets. Retained claims also still reflect the underlying LLM's knowledge limitations and biases.

## Limitations

While DCF achieves strong empirical results, we identify several limitations:

**Quantile Instability at Strict Thresholds.** At very low $\alpha$, collapse is primarily a data-size limitation: after train/validation/test and calibration splits, the calibration subset may be too small to support quantile estimation at 0.01–0.02 resolution. To remain conservative, our implementation defaults to the minimum threshold $\tau_{\min}$ when the target quantile is under-supported, which yields an empty prediction set (retain none) for that $\alpha$. This behavior is pronounced on MATH at $\alpha \leq 0.02$. On FELM, it does not collapse at $\alpha = 0.01$ (17.9% retention; Table 9), though strict thresholds still show sensitivity (e.g., a slight target miss at $\alpha = 0.02$).

**Reduced Gains When Frequency Is Discriminative.** DCF's advantage is largest when self-consistency frequency (or any baseline feature) poorly discriminates valid from invalid claims—as occurs with long reasoning chains, uncommon but valid techniques, or semantically incoherent outputs. On FELM, where simpler reasoning (4.0 vs. 7.3 claims per problem) makes frequency more reliable, DCF's gains are smaller and frequency-based baselines occasionally outperform DCF at mid-range $\alpha$ (Table 9).

**MATH Dataset Scale and Annotation.** Our main MATH evaluation uses 202 problems with manually annotated reasoning graphs. We additionally validated DCF on a $2\times$ expanded MATH-407 dataset (Section E.3.1); results are qualitatively consistent with the main paper. Our operationalization of coherent factuality treats fully hallucinated ADGs as incoherent, which influences graph metrics like `nx_reachability`. The learned combination may not generalize to settings with well-structured but incorrect reasoning, where content-based features may become more critical.

**Scorer Architecture.** Our main results use a linear scorer for interpretability. A complementary case study at $\alpha = 0.09$ on MATH-407 (Section E.3.2) shows that a shallow MLP recovers an additional $+11.5\%$ retention over the linear scorer when calibration data is sufficient, indicating that scorer capacity is a straightforward axis of further improvement. We retain the linear scorer as the primary model to enable the interpretability analysis in Section 4.4.

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

# A. Proofs

## A.1. Statement and Proof of Theorem 3.1

Fix ADG $\mathcal{G} = (V, E)$, false claim nodes $V^- \subseteq V$, scorer $\pi_\theta$, threshold grid $\mathcal{T}$, and graph family $\mathcal{U}_\mathcal{T}$. Let $p_{v,\tau} = \sigma\left((\tau - r_v + \sqrt{T_p})/T_p\right)$ with[8] $\sup(\lambda \mathcal{T}) - \inf(\lambda \mathcal{T}) \leq 1$. Then:

$$\lim_{\beta \to \infty} \lim_{\tau_s \to 0} \lim_{T_p \to 0^+} \tilde{\tau} = \nu(X, Y, \mathcal{U}_\mathcal{T}) \tag{10}$$

where $\nu(X, Y, \mathcal{U}_\mathcal{T})$ is the hard nonconformity score from CF.[9] Setting $\epsilon = T_p$, $\tau_s = T_p^s$, and $\beta = T_p^a$ for $s \in (0, 1), a < 0$ yields single-limit convergence.

*Proof.* For simplicity of notation set $T = T_p$. Checking limits, note as $T \to 0^+$, $p_{v,\tau} \longrightarrow \Theta(\tau - r_v)$ pointwise, where $\Theta(x)$ is the Heaviside step function with $\Theta(0) = 1$.

*Case 1.* $r_v > \tau$ and vertex $v$ is not coherently factual. Then

$$r_v > \tau \Rightarrow \lim_{T \to 0^+} p_{v,\tau} = 0. \tag{11}$$

*Case 2.* $r_v \leq \tau$ and vertex $v$ is coherently factual. Then

$$r_v \leq \tau \Rightarrow \lim_{T \to 0^+} p_{v,\tau} = 1. \tag{12}$$

After taking an ancestral geometric mean, $q_{v,\tau} = \left(\Pi_{u \in \text{Anc}(v) \cup \{v\}} p_{u,\tau}^{w_u}\right)^{1/(|\text{Anc}(v)|+1)}$ is sent to 0 if any of $v$ or $v$'s ancestors had risk score greater than $\tau$, and is sent to 1 otherwise. Then

$$\lim_{T \to 0^+} Q_\tau = \begin{cases} 0 & \text{for} \quad \exists v \in V^- \,|\, \forall u \in \text{Anc}(v) \cup \{v\}, r_u \leq \tau \\ 1 & \text{for} \quad \forall v \in V^-, \exists u \in \text{Anc}(v) \cup \{v\} \,|\, r_u > \tau \end{cases}, \tag{13}$$

and $\tau_s$ has no influence. Now there are two cases to consider for violation score: $V^-$ can be empty or nonempty. If nonempty, then the limit of $Q_{\tau_{\max}}$ is zero, so

$$\lim_{T \to 0^+} V_\tau = 1 - \frac{\lim Q_\tau^{1/\tau_s} - 0}{1 - 0 + \lim T} = 1 - \lim_{T \to 0^+} Q_\tau^{1/\tau_s}.$$

Now if $V^-$ is empty, then all $Q_\tau$ approach 1, and by L'Hôpital's rule,

$$\lim_{T \to 0^+} V_\tau = 1 - \frac{0 - 0}{0 - 0 + 1} = 1.$$

Then violation has that

$$\lim_{T \to 0^+} V_\tau = \begin{cases} 1 & \text{for} \quad U_\tau \text{ is not Coherently Factual or } V^- = \emptyset \\ 0 & \text{for} \quad U_\tau \text{ is Coherently Factual} \end{cases}. \tag{14}$$

In the limit $\beta \to \infty$, $w_{\tau_i}^{\text{cal}} = \mathbb{1}\{s_{\tau_i} = \sup\{s_{\tau_j} \,|\, \tau_j \in \mathcal{T}\}\}$. Note $\sup(\lambda \mathcal{T}) - \inf(\lambda \mathcal{T}) \leq 1$. In the case $V^- \neq \emptyset$, if any subgraph $U_{\tau_i}$ is not CF, then $V_{\tau_i} = 1$, so the maximum $s_{\tau_i}$ corresponds to the greatest $\lambda \tau_i$ such that $V_{\tau_i} = 0$ and $U_{\tau_i}$ is CF. If $V^- = \emptyset$, then the maximum $s_{\tau_i}$ corresponds to $\lambda \tau_{\max}$, permitting all claims, as we wanted. We lastly need the fact that

$$\forall \tau' \in \mathcal{T} \,|\, \tau' \leq \tau, U_\tau \text{ Coherently Factual} \Rightarrow U_{\tau'} \text{ Coherently Factual}, \tag{15}$$

which holds since $Q_\tau = 1$ and $\tau' \leq \tau$ implies $Q_{\tau'} = 1$. Therefore

$$\lim_{\beta \to \infty} \lim_{\tau_s \to 0^+} \lim_{T \to 0^+} \tilde{\tau} = \nu(X, Y, \mathcal{U}_\mathcal{T}) \tag{16}$$

---

[8]In practice, we let $\lambda$ vary as a hyperparameter. The theorem's condition could be guaranteed by making $\lambda$ differentiably approximate $1/(\sup(\mathcal{T}) - \inf(\mathcal{T}))$, then applying a limit to recover the hard nonconformity score.

[9]The theorem requires a $\sqrt{T_p}$ margin in (1), but in practice a margin of 0 yields similar soft keep probabilities.

It remains to show that setting $\tau_s = T^s$ and $\beta = T^a$ for $s \in (0,1), a < 0$ yields single-limit convergence. Note that $\beta(T) \to \infty$ and $\tau_s \to 0$ as $T \to 0$, let $s_{\max}(T) = \arg\max_{s \in \{s_\tau\}_{\tau \in \mathcal{T}}} \lim_{T \to 0^+} s_\tau$, and let $n_{\max}$ be the number of $s_\tau$ with the same limit as $s_{\max}$. Then for $\tau_{\sup}$ corresponding to $s_{\max}$,

$$\lim_{T \to 0^+} \tilde{\tau} = \lim_{T \to 0^+} \frac{\sum_{\tau \in \mathcal{T}} \exp(\beta s_\tau)\tau}{\sum_{\tau \in \mathcal{T}} \exp(\beta s_\tau)} = \lim_{T \to 0^+} \frac{\sum_{\tau \in \mathcal{T}} \exp(\beta s_\tau - \beta s_{\max})\tau}{\sum_{\tau \in \mathcal{T}} \exp(\beta s_\tau - \beta s_{\max})} = \frac{\tau_{\sup} n_{\max}}{n_{\max}} = \tau_{\sup},$$

if $s_{\max}$ is well defined, which is seen through convergence of $Q_\tau^{1/\tau_s}$. Now it remains to show that

$$\lim_{T \to 0^+} Q_\tau^{1/\tau_s} = \begin{cases} 0 & \text{for} \quad \exists v \in V^- \,|\, \forall u \in \text{Anc}(v) \cup \{v\}, r_u \leq \tau \\ 1 & \text{for} \quad \forall v \in V^-, \exists u \in \text{Anc}(v) \cup \{v\} \,|\, r_u > \tau \end{cases}, \tag{17}$$

similar to equation 13 but with an exponent that could prevent convergence to 1. We first show that in the case where $Q_\tau \to 1$, we also have $Q_\tau^{1/\tau_s} \to 1$. So $\forall v \in V^-, \exists u \in \text{Anc}(v) \cup \{v\} \,|\, r_u > \tau$. We can write

$$\lim_{T \to 0^+} Q_\tau^{|V^-|/\tau_s} = \Pi_{v \in V^-} \lim_{T \to 0^+} (1 - q_{v,\tau})^{1/\tau_s}.$$

Set $L_v = \lim_{T \to 0^+} (1 - q_{v,\tau})^{1/\tau_s}$. We want $L_v \to 1$ for all $v \in V^-$. Notice we can express $L_v$ in the following manner:

$$L_v = \lim_{T \to 0^+} (1 - f_1(T)f_2(T)...)^{1/\tau_s}, \tag{18}$$

where each $f_i = p_{u,\tau}^{c_u}$ with appropriate constant $c_u > 0$. At least one $f_i$ has $\lim_{T \to 0^+} f_i(T) = 0$ and others approach 1 or 0. Now

$$L_v = \exp \lim_{T \to 0^+} \ln(1 - f_1(T)f_2(T)...)/\tau_s = \exp \lim_{T \to 0^+} \frac{-1}{1 - f_1(T)f_2(T)...} (\tau_s')^{-1} \sum_i f_1(T)f_2(T)...\hat{f}_i(T)...f_n(T)f_i(T)'$$

by L'Hôpital's rule, where the hat indicates omission of that term. Now $\frac{-1}{1 - f_1(T)f_2(T)...} \to -1$, and $\tau_s' = \frac{1}{s}T^{s-1} \to \infty$ so $(\tau_s')^{-1} \to 0$. The remaining terms are:

$$\sum_i f_1(T)f_2(T)...\hat{f}_i(T)...f_n(T)f_i(T)'.$$

Each $f_i$ approaches 1 or 0, so we must check $f_i(T)'$ for convergence. Explicitly, for an $f_i$ there is some $u \in \{v\} \cup \text{Anc}(v)$ such that

$$f_i(T) = p_{u,\tau}^{w_u/(1+\gamma|\text{Anc}(v)|)} = \frac{1}{(1 + \exp((r_u - \tau - \sqrt{T})/T))^{c_2}} = \frac{1}{(1 + \exp(c_1/T - 1/\sqrt{T}))^{c_2}},$$

with $c_2 = \frac{w_u}{1+\gamma|\text{Anc}(v)|} \in (0,1)$ and $c_1 = r_u - \tau$. After differentiation and taking the limit, we find

$$\lim_{T \to 0^+} f_i(T)' = 0.$$

Thus $L_v = \exp(0) = 1$ for each $v \in V^-$ when $\forall v \in V^-, \exists u \in \text{Anc}(v) \cup \{v\} \,|\, r_u > \tau$. Then $\lim_{T \to 0^+} Q_\tau^{1/\tau_s} = 1$.

The remaining case to prove is when $\exists v \in V^- \,|\, \forall u \in \text{Anc}(v) \cup \{v\}, r_u \leq \tau$. Here, there exists a $v \in V^-$ such that each $f_i$ in equation 18 approaches 1, and thus $L_v$ approaches 0 and $\lim_{T \to 0^+} Q_\tau^{1/\tau_s} = 0$. This completes the proof of equation 17. The deductions following equation 13 follow. Thus setting $\tau_s = T^s$ and $\beta = T^a$ for $s \in (0,1), a < 0$ yields

$$\lim_{T \to 0^+} \tilde{\tau} = \nu(X, Y, \mathcal{U}_\mathcal{T}). \tag{19}$$

$\square$

### A.2. Statement and Proof of Theorem 3.2

Fix graph $\mathcal{G} = (V, E)$, risk scores $\{r_v\}_{v \in V}$, and calibrated threshold $\tau_\alpha$. Suppose $\{\tau \in \mathcal{T} \mid \tau < \tau_\alpha\} \neq \emptyset$ and $\forall u \in V, w_u \neq 0$. Let[10] $w_\tau^{(\text{unnorm})} = \exp(\beta \cdot \tau) \cdot \sigma \left( \frac{\tau_\alpha - \tau - \sqrt{\tau_z}}{\tau_z} \right)$. Then:

$$\{v \in V \mid \lim_{T_p \to 0^+} \lim_{\beta \to \infty} \lim_{\tau_z \to 0^+} q_v \in [0.5, 1]\} = U_{\text{filtered}}, \tag{20}$$

where $U_{\text{filtered}}$ is the CF prediction.[11] Setting $\tau_z = T_p^{ab}$, $\beta = 1/T_p^a$ for $a > 0$, $b > 2$ yields single-limit convergence.

*Proof.* For simplicity of notation let $T = T_p$ and $\hat{\tau}_\alpha = \tau_\alpha$. Since

$$\lim_{T \to 0^+} p_{v,\tau} = \begin{cases} 1 & \text{for} \quad r_v < \tau \\ \frac{1}{2} & \text{for} \quad r_v = \tau \\ 0 & \text{for} \quad r_v > \tau \end{cases}, \tag{21}$$

we have

$$\lim_{T \to 0^+} q_{v,\tau} \in \begin{cases} \{0\} & \text{if} \quad \exists u \in \text{Anc}(v) \cup \{v\} : r_u > \tau \\ [\frac{1}{2}, 1] & \text{if} \quad \forall u \in \text{Anc}(v) \cup \{v\}, r_u \leq \tau \end{cases}. \tag{22}$$

Note

$$\lim_{\tau_z \to 0^+} w_\tau^{(\text{unnorm})} = \begin{cases} 0 & \text{for} \quad \tau \geq \tau_\alpha \\ \exp(\beta\tau) & \text{for} \quad \tau < \tau_\alpha \end{cases}. \tag{23}$$

Now as $\beta \to \infty$, $w_\tau \to \mathbb{1}\{\tau = \sup\{\tau \in \mathcal{T} \mid \tau < \tau_\alpha\}\}/n_j$, where $n_j = |\{\tau \in \mathcal{T} \mid \tau = \sup\{\tau \in \mathcal{T} \mid \tau < \tau_\alpha\}\}|$. Since identical $\tau$ give identical coefficients $q_{v,\tau}$ for each $w_\tau$ in equation 8, we find that

$$\lim_{\beta \to \infty} \lim_{\tau_z \to 0^+} q_v = q_{v,\tau} : \tau = \sup\{\tau \in \mathcal{T} \mid \tau < \tau_\alpha\} \tag{24}$$

Then $\{v \in V \mid \lim_{T \to 0^+} \lim_{\beta \to \infty} \lim_{\tau_z \to 0^+} q_v \in [0.5, 1]\} = \{v \in V \mid \forall u \in \text{Anc}(v) \cup \{v\}, r_u \leq \sup\{\tau \in \mathcal{T} \mid \tau < \tau_\alpha\}\}$, which is equivalent to $U_{\text{filtered}}$.

It remains to show that setting $\beta = 1/T^a$ and $\tau_z = T^{ab}$ for $a > 0, b > 1$ yields single-limit convergence. We may write

$$q_v = \frac{\sum_\tau \exp(\tau/T^a)\sigma(\frac{\tau_\alpha - \tau - T^{ab/2}}{T^{ab}})q_{v,\tau}}{\sum_\tau \exp(\tau/T^a)\sigma(\frac{\tau_\alpha - \tau - T^{ab/2}}{T^{ab}})}. \tag{25}$$

Let $\tau_j = \sup\{\tau \in \mathcal{T} \mid \tau < \tau_\alpha\}$ so

$$q_v = \frac{\sum_\tau \exp(\tau/T^a - \tau_j/T^a)\sigma(\frac{\tau_\alpha - \tau - T^{ab/2}}{T^{ab}})q_{v,\tau}}{\sum_\tau \exp(\tau/T^a - \tau_j/T^a)\sigma(\frac{\tau_\alpha - \tau - T^{ab/2}}{T^{ab}})}. \tag{26}$$

Only terms matching $\tau_j$ survive in both sums. Terms with $\tau < \tau_j$ approach 0 via the first exponential, and terms with $\tau > \tau_j$ also have $\tau \geq \tau_\alpha$, so

$$\lim_{T \to 0^+} \exp(\frac{\tau - \tau_j}{T^a})\sigma(\frac{\tau_\alpha - \tau - T^{ab/2}}{T^{ab}}) = \lim_{T \to 0^+} \frac{1}{e^{(\tau_j - \tau)/T^a} + \exp(\frac{(\tau - \tau_\alpha) + (\tau_j - \tau)T^b + T^{ab/2}}{T^{ab}})} = 0 \tag{27}$$

since $b > 2$ and $a > 0$. Then

$$\lim_{T \to 0^+} q_v = \frac{\sum_\tau \mathbb{1}\{\tau = \tau_j\} \lim_{T \to 0^+} q_{v,\tau}}{\sum_\tau \mathbb{1}\{\tau = \tau_j\}} = \lim_{T \to 0^+} q_{v,\tau_j}. \tag{28}$$

Since equation 22 holds in this case and $\tau_j$ is identical to the $\tau$ of equation 24, we have

$$\{v \in V \mid \lim_{T \to 0^+} q_v \in [0.5, 1]\} = U_{\text{filtered}} \tag{29}$$

$\square$

---

[10]This requires a $\sqrt{\tau_z}$ margin in equation 7. In practice, this was left at zero.

[11]This theorem uses equation (1) in its original form; extending it to support the margin from Theorem 3.1 is straightforward.

# B. Algorithms

## B.1. Differentiable Calibration

---

**Algorithm 1** Differentiable Calibration

---

1: **Input:** Calibration set $\mathcal{D}_{\text{cal}} = \{(\mathcal{G}_i, \{x_v^{(i)}\}, \{y_v^{(i)}\})\}_{i=1}^n$, scorer $\pi_\theta$, target miscoverage $\alpha$, hyperparameters $C, T_p, \beta, \gamma, \lambda, \tau_s, \rho, \epsilon, m \in \mathbb{R}$
2: **Output:** Calibrated threshold $\hat{\tau}_\alpha$
3: **for** each graph $(\mathcal{G}, \{x_v\}, \{y_v\})$ in $\mathcal{D}_{\text{cal}}$ **do**
4:     **Step 1: Scoring and risk.**
5:     **for** $v \in V$ **do**
6:         $r_v = C - \pi_\theta(x_v)$
7:     **end for**
8:     **Step 2: Build $\tau$-grid.**
9:     $\mathcal{T} = \{\min_v r_v - m, \ \{r_v\}_{v \in V}, \ \max_v r_v + m\}$ (sorted)
10:     **Step 3: Soft keep.**
11:     **for** $(v, \tau) \in V \times \mathcal{T}$ **do**
12:         $p_{v,\tau} = \sigma\left(\frac{\tau - r_v}{T_p}\right)$
13:     **end for**
14:     **Step 4: Ancestor coherence.**
15:     **for** $(v, \tau) \in V \times \mathcal{T}$ **do**
16:         $w_u = \begin{cases} \gamma & \text{if } u \in \text{Anc}(v) \\ 1 & \text{if } u = v \end{cases}$ for all $u \in \text{Anc}(v) \cup \{v\}$
17:         $\log q_{v,\tau} = \frac{\sum_{u \in \text{Anc}(v) \cup \{v\}} w_u \cdot \log(p_{u,\tau} + \epsilon)}{\sum_{u \in \text{Anc}(v) \cup \{v\}} w_u}$
18:     **end for**
19:     **Step 5: Validity on negatives.**
20:     $V^- = \{v \in V : y_v = 0\}$
21:     **for** $\tau \in \mathcal{T}$ **do**
22:         $\log Q_\tau = \frac{1}{|V^-|} \sum_{v \in V^-} \log(1 - \exp(\log q_{v,\tau}) + \epsilon)$
23:     **end for**
24:     **Step 6: Violation and soft supremum.**
25:     **for** $\tau \in \mathcal{T}$ **do**
26:         $V_\tau = 1 - \exp(\log(Q_\tau)/\tau_s)$                                  {Implementation uses min-max variant; see footnote on Eq. 4}
27:     **end for**
28:     Normalize: $\hat{\tau}_{\text{norm}} = \text{MinMax}(\mathcal{T})$, $\hat{V} = \text{MinMax}(\{V_\tau\})$                     {Scale to $[0, 1]$}
29:     **for** $\tau \in \mathcal{T}$ **do**
30:         $s_\tau = \hat{\tau}_{\text{norm}} - \lambda \cdot \hat{V}_\tau$
31:         $w_\tau^{\text{cal}} = \frac{\exp(\beta s_\tau)}{\sum_{\tau' \in \mathcal{T}} \exp(\beta s_{\tau'})}$
32:     **end for**
33:     $\tilde{\tau} = \sum_{\tau \in \mathcal{T}} w_\tau^{\text{cal}} \cdot \tau$
34:     Store $\tilde{\tau}$
35: **end for**
36: **Calibration via soft quantile:**
37: $q = \frac{\lceil (n+1)(1-\alpha) \rceil}{n}$
38: $\hat{\tau}_\alpha = \text{SoftQuantile}(\{\tilde{\tau}_1, \dots, \tilde{\tau}_n\}, q, \rho)$
39: **return** $\hat{\tau}_\alpha$

---

## B.2. Differentiable Prediction

---

**Algorithm 2** Differentiable Prediction

---

1: **Input:** Test set $\mathcal{D}_{\text{test}} = \{(\mathcal{G}_i, \{x_v^{(i)}\})\}_{i=1}^M$, scorer $\pi_\theta$, calibrated threshold $\hat{\tau}_\alpha$, hyperparameters $C, T_p, \beta, \gamma, \tau_z, \epsilon, m \in \mathbb{R}$
2: **Output:** Soft retention probabilities $\{\mathbf{q}^{(i)}\}_{i=1}^M$ where $\mathbf{q}^{(i)} \in [0, 1]^{|V_i|}$
3: **for** each graph $(\mathcal{G}, \{x_v\})$ in $\mathcal{D}_{\text{test}}$ **do**
4:     **Step 1: Scoring and risk.**
5:     **for** $v \in V$ **do**
6:         $r_v = C - \pi_\theta(x_v)$
7:     **end for**
8:     **Step 2: Build $\tau$-grid.**
9:     $\mathcal{T} = \{\min_v r_v - m, \ \{r_v\}_{v \in V}, \ \max_v r_v + m\}$ (sorted)
10:     **Step 3: Soft keep.**
11:     **for** $(v, \tau) \in V \times \mathcal{T}$ **do**
12:         $p_{v,\tau} = \sigma\left(\frac{\tau - r_v}{T_p}\right)$
13:     **end for**
14:     **Step 4: Ancestor coherence.**
15:     **for** $(v, \tau) \in V \times \mathcal{T}$ **do**
16:         $w_u = \begin{cases} \gamma & \text{if } u \in \text{Anc}(v) \\ 1 & \text{if } u = v \end{cases}$ for all $u \in \text{Anc}(v) \cup \{v\}$
17:         $\log q_{v,\tau} = \frac{\sum_{u \in \text{Anc}(v) \cup \{v\}} w_u \cdot \log(p_{u,\tau} + \epsilon)}{\sum_{u \in \text{Anc}(v) \cup \{v\}} w_u}$
18:     **end for**
19:     **Step 5: Soft gated argmax.**
20:     **for** $\tau \in \mathcal{T}$ **do**
21:         $w_\tau^{(\text{unnorm})} = \exp(\beta \cdot \tau) \cdot \sigma\left(\frac{\hat{\tau}_\alpha - \tau}{\tau_z}\right)$
22:     **end for**
23:     Normalize: $w_\tau = \frac{w_\tau^{(\text{unnorm})}}{\sum_{\tau' \in \mathcal{T}} w_{\tau'}^{(\text{unnorm})}}$
24:     **Step 6: Weighted combination.**
25:     **for** $v \in V$ **do**
26:         $q_v = \sum_{\tau \in \mathcal{T}} w_\tau \cdot q_{v,\tau}$
27:     **end for**
28:     Store $\mathbf{q} = (q_v)_{v \in V}$
29: **end for**
30: **return** $\{\mathbf{q}^{(1)}, \ldots, \mathbf{q}^{(M)}\}$

---

## B.3. Training Procedure

---

**Algorithm 3** Training $\pi_\theta$

---

1: **Input:** Dataset $\mathcal{D} = \{(\mathcal{G}_i, \{x_v^{(i)}\}, \{y_v^{(i)}\})\}_{i=1}^N$, scorer $\pi_\theta$, miscoverage $\alpha$, learning rate $\eta$, epochs $E$, hyperparameters $C, T_p, \beta, \gamma, \lambda, \tau_s, \rho, \epsilon, m$
2: **Output:** Trained scorer $\pi_\theta$
3: **Step 1: Data split.**
4: Partition $\mathcal{D}$ into train $\mathcal{D}_{\text{train}}$ and validation $\mathcal{D}_{\text{val}}$ sets
5: **Step 2: Initialize.**
6: Initialize scorer parameters $\theta$ and Adam optimizer with learning rate $\eta$
7: **for** epoch $e = 1, \ldots, E$ **do**
8:     **Step 3: Training.**
9:     Split $\mathcal{D}_{\text{train}}$ into calibration $\mathcal{D}_{\text{cal}}$ and prediction $\mathcal{D}_{\text{pred}}$ subsets
10:     Calibrate: $\hat{\tau}_\alpha \leftarrow \text{DifferentiableCalibration}(\mathcal{D}_{\text{cal}}, \pi_\theta, \alpha, \ldots)$ (Algorithm 1)
11:     Predict: $\{\mathbf{p}^{(i)}\} \leftarrow \text{DifferentiablePrediction}(\mathcal{D}_{\text{pred}}, \pi_\theta, \hat{\tau}_\alpha, \ldots)$ (Algorithm 2)
12:     Compute loss: $\mathcal{L} = -\frac{1}{|\mathcal{D}_{\text{pred}}|} \sum_{i \in \mathcal{D}_{\text{pred}}} \sum_{v \in V_i} p_v^{(i)} \cdot y_v^{(i)}$
13:     Backpropagate: $\nabla_\theta \mathcal{L}$ and update $\theta \leftarrow \theta - \eta \cdot \nabla_\theta \mathcal{L}$
14: **end for**
15: **return** Trained scorer $\pi_\theta$

---

**Problem:** The endpoints of a segment are $(1, 4)$ and $(1, 10)$. What is the sum of the coordinates of the midpoint?

**Answer:** 8

**Claims:**

0: Midpoint formula
1: $\frac{1+1}{2} = 1$
2: $\frac{4+10}{2} = 7$
3: Midpoint is $(1, 7)$
4: Sum: $1 + 7 = 8$

*Figure 6.* Example ADG for a MATH problem. Nodes represent atomic claims; edges indicate logical dependencies.

## C. Figures

### C.1. Training Flow

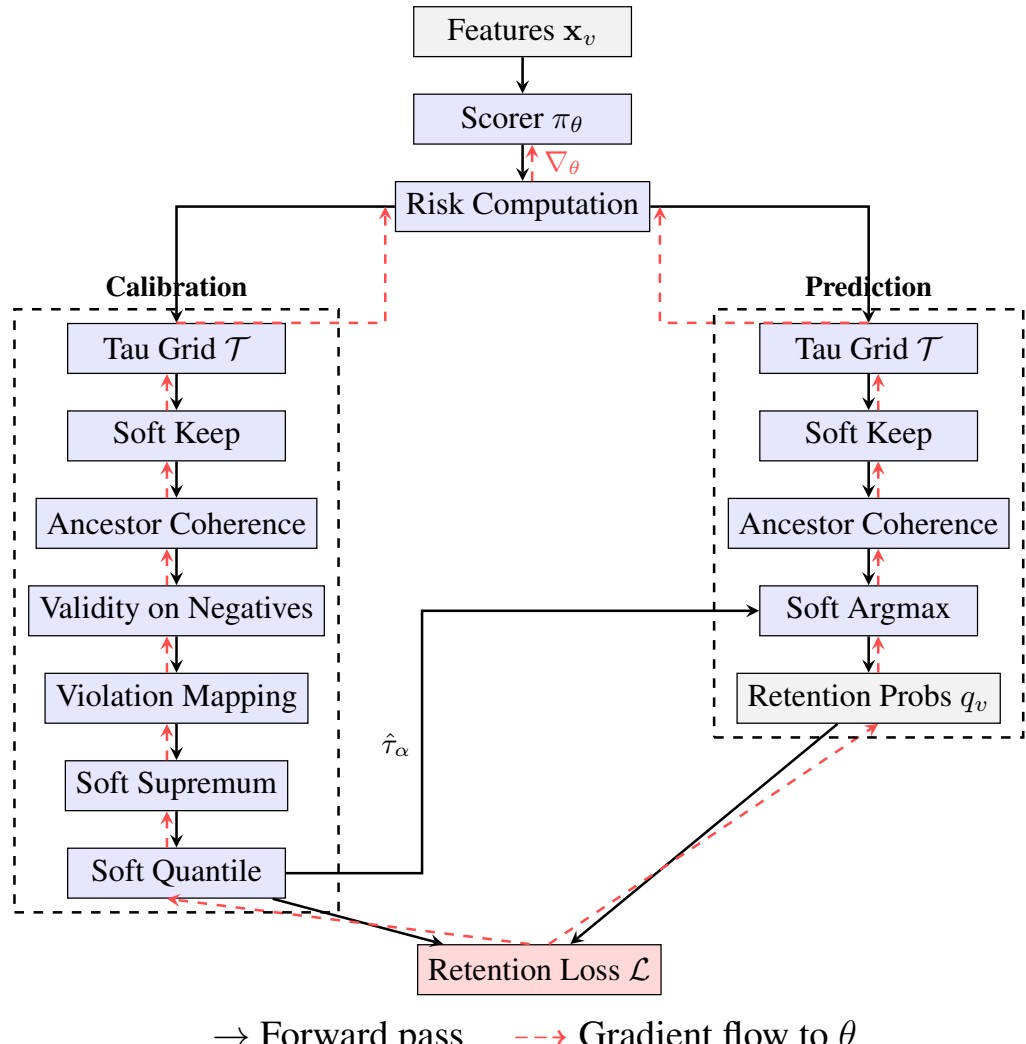

$\rightarrow$ Forward pass $\quad --\rightarrow$ Gradient flow to $\theta$

*Figure 5.* Gradient flow through DCF training.

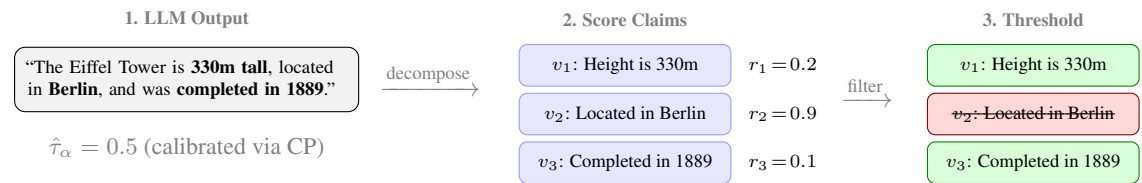

*Figure 7.* LLM conformal factuality. An LLM response is decomposed into atomic subclaims, each assigned a risk score. The threshold $\hat{\tau}_\alpha$ is calibrated via CP to guarantee a $1 - \alpha$ factuality rate. Claims exceeding this threshold are removed (e.g., the hallucinated "Berlin").

### C.2. Example Approximate Deducibility Graph

### C.3. Conformal Factuality Pipeline

## D. Experimental Details

### D.1. Feature Descriptions

Each claim has 31 features organized into four categories:

**Base Scoring (1 feature).** `frequency-score`: Self-consistency frequency across multiple LLM generations.

**Semantic Coherence (8 features).** `claim_index` (position in chain), `coherent_to_ancestors` (logical consistency), `inference_gap_size` (missing steps), `has_missing_dependencies`, `has_unnecessary_dependencies`, `problem_relevance`, `uses_problem_data`, and additional validation flags.

**Domain Indicators (12 features).** Binary flags for mathematical topics: `linear_equations`, `quadratic_equations`, `polynomial_algebra`, `rational_expressions`, `exponents_logarithms`, `inequalities`, `sequences_series`, `functions`, `word_problems`, `coordinate_geometry`, `abstract_algebra`, `complex_numbers`.

**Graph Metrics (10 features).** NetworkX-computed properties: `nx_in_degree`, `nx_out_degree`, `nx_pagerank`, `nx_betweenness`, `nx_closeness`, `nx_clustering`, `nx_is_source`, `nx_is_sink`, `nx_reachability`, `nx_depth_from_sources`.

**FELM Feature Selection.** For FELM, we exclude MATH-specific domain indicators and use two feature configurations based on $\alpha$:

**7 features** (for $\alpha \leq 0.08$): `frequency-score`, `coherent_to_ancestors`, `inference_gap_size`, `has_missing_dependencies`, `nx_pagerank`, `nx_reachability`, `nx_out_degree`.

**20 features** (for $\alpha \geq 0.09$): The 7-feature set plus `claim_index`, `has_unnecessary_dependencies`, `problem_relevance`, `uses_problem_data`, `domain_science`, `domain_reasoning`, `nx_in_degree`, `nx_betweenness`, `nx_closeness`, `nx_clustering`, `nx_is_source`, `nx_is_sink`, `nx_depth_from_sources`.

### D.2. Training Configuration

Both MATH and FELM experiments share the following training setup:

- **Model**: LogisticClaimScorer—30 input features for MATH, 7 or 20 for FELM depending on $\alpha$ (see Section D.1)
- **Optimizer**: Adam with learning rate 0.015, no weight-decay regularization
- **Training**: Up to 100 epochs with early stopping (patience $= 10$)
- **Evaluation**: 20-fold cross-validation with 70/15/15 train/validation/test splits
- **Hyperparameter selection**: Grid search over all $\alpha$ values

**Validation Hyperparameters.** The surrogate-fidelity experiments in Section 4.2 use fixed hyperparameters: $T_p = 0.01$, $\beta = 1.0$, $\gamma = 1.0$, $\lambda = 1.0$, $\tau_s = 0.001$, $\tau_z = 0.001$, $m = 20.0$. These experiments use the exponential violation

transformation (Eq. 4) with $\tau_s \to 0$ to validate convergence to hard CF; optimization experiments (RQ2 onwards) use the min-max variant described in the footnote of Eq. 4.

## D.3. Optimized Hyperparameters

Tables 5 and 6 list the per-$\alpha$ hyperparameters selected by grid search for DCF.

*Table 5.* DCF hyperparameters for MATH dataset selected via grid search.

| $\alpha$ | $\gamma$ | $\lambda$ | $T_p$ | lr |
|---|---|---|---|---|
| 0.03 | 4.0 | 1.35 | 1.0 | 0.015 |
| 0.04 | 5.0 | 1.30 | 0.1 | 0.015 |
| 0.05 | 6.0 | 1.60 | 0.1 | 0.015 |
| 0.06 | 6.0 | 1.60 | 0.1 | 0.015 |
| 0.07 | 1.5 | 1.70 | 0.1 | 0.015 |
| 0.08 | 6.0 | 1.75 | 0.1 | 0.015 |
| 0.09 | 2.0 | 1.70 | 0.1 | 0.015 |
| 0.10 | 5.0 | 1.90 | 0.1 | 0.015 |

*Table 6.* DCF hyperparameters for FELM dataset selected via grid search. All configurations share $\beta = 1$ and lr $= 0.015$.

| $\alpha$ | $\gamma$ | $\lambda$ | $T_p$ |
|---|---|---|---|
| 0.01 | 1.50 | 1.20 | 1.0 |
| 0.02 | 1.00 | 1.50 | 1.0 |
| 0.03 | 1.25 | 1.30 | 1.0 |
| 0.04 | 1.50 | 1.45 | 0.5 |
| 0.05 | 1.25 | 1.45 | 0.5 |
| 0.06 | 1.00 | 1.00 | 1.0 |
| 0.07 | 1.25 | 0.90 | 1.0 |
| 0.08 | 1.25 | 1.50 | 1.0 |
| 0.09 | 1.00 | 1.65 | 1.0 |
| 0.10 | 1.25 | 1.90 | 0.5 |

## D.4. Baseline $\beta_{\text{mix}}$ Optimization

For fair comparison, we optimized $\beta_{\text{mix}}$ for the frequency-based CF baseline via grid search over $\{0.0, 0.1, \ldots, 1.0\}$. Table 7 reports the selected values for each dataset. We use these optimized values in all comparisons to ensure DCF is evaluated against the strongest possible baseline.

*Table 7.* Optimized $\beta_{\text{mix}}$ for the CF baseline on each dataset.

| $\alpha$ | MATH | FELM |
|---|---|---|
| 0.01 | 0.0 | 0.8 |
| 0.02 | 0.0 | 0.6 |
| 0.03 | 0.8 | 0.5 |
| 0.04 | 0.0 | 0.2 |
| 0.05 | 0.0 | 0.4 |
| 0.06 | 0.0 | 0.3 |
| 0.07 | 0.9 | 0.4 |
| 0.08 | 0.8 | 0.1 |
| 0.09 | 0.2 | 0.1 |
| 0.10 | 0.8 | 0.0 |

*Table 8.* MATH dataset results: DCF vs. CF baseline across $\alpha$ values. Coverage target is $1-\alpha$. Bold coverage indicates DCF meets the target. Retention is the mean number of claims retained per problem; $\Delta$ Ret shows percentage improvement of DCF over CF.

| $\alpha$ | Coverage (%) | | Retention | | | Meets |
|---|---|---|---|---|---|---|
| | DCF | CF | DCF | CF | $\Delta$ (%) | Target? |
| 0.03 | 96.55 | 97.09 | 1.76 | 0.73 | +141.1 | No* |
| 0.04 | 95.59 | 95.80 | 2.22 | 1.14 | +94.7 | No* |
| 0.05 | **95.55** | 94.85 | 2.31 | 1.44 | +60.4 | Yes |
| 0.06 | **94.14** | 94.10 | 3.56 | 1.74 | +104.6 | Yes |
| 0.07 | **93.14** | 93.00 | 3.63 | 2.01 | +80.6 | Yes |
| 0.08 | **94.14** | 91.82 | 3.69 | 2.31 | +59.7 | Yes |
| 0.09 | 90.64 | 90.59 | 5.57 | 2.75 | +102.5 | No* |
| 0.10 | 89.64 | 89.68 | 5.99 | 3.33 | +79.9 | No* |

*Near-miss: all misses are within 0.5pp of target.

*Table 9.* FELM dataset results: DCF vs. CF baseline across $\alpha$ values. Coverage target is $1-\alpha$. Bold coverage indicates DCF meets the target. Retention is the percentage of claims retained; $\Delta$ Ret shows relative percentage improvement of DCF over CF.

| $\alpha$ | Coverage (%) | | Retention (%) | | | Meets |
|---|---|---|---|---|---|---|
| | DCF | CF | DCF | CF | $\Delta$ (%) | Target? |
| 0.01 | **99.15** | 99.19 | 17.9 | 11.1 | +61.3 | Yes |
| 0.02 | 97.75 | 98.08 | 35.7 | 22.4 | +59.4 | No* |
| 0.03 | **97.32** | 97.11 | 36.4 | 29.5 | +23.4 | Yes |
| 0.04 | **96.47** | 95.89 | 42.0 | 36.2 | +16.0 | Yes |
| 0.05 | **95.35** | 95.10 | 46.3 | 41.5 | +11.6 | Yes |
| 0.06 | **94.92** | 93.94 | 48.1 | 49.2 | −2.2 | Yes† |
| 0.07 | **94.21** | 93.13 | 49.6 | 55.0 | −9.8 | Yes† |
| 0.08 | **93.64** | 92.15 | 52.7 | 61.3 | −14.0 | Yes† |
| 0.09 | **91.39** | 90.80 | 69.0 | 66.4 | +3.9 | Yes |
| 0.10 | **90.54** | 90.08 | 71.2 | 68.9 | +3.3 | Yes |

*DCF misses coverage target at $\alpha = 0.02$ (97.75% < 98.0%).

†DCF meets coverage but CF achieves higher retention.

*Table 10.* MATH-407 (Section E.3.1) results: DCF (Logistic) vs. frequency-CF baseline across $\alpha$ values. Coverage target is $1-\alpha$. Bold coverage indicates DCF meets the target. Frequency-CF uses per-$\alpha$ optimized $\beta_{\mathrm{mix}}$. Retention is mean claims retained per problem; $\Delta$ Ret is the percentage improvement of DCF over CF. The DCF entry at $\alpha = 0.09$ uses the wider hyperparameter sweep from Section E.3.2.

| $\alpha$ | Coverage (%) | | Retention | | | Meets |
|---|---|---|---|---|---|---|
| | DCF | CF | DCF | CF | $\Delta$ (%) | Target? |
| 0.01 | **100.00** | 99.26 | 0.00 | 0.17 | −100 | Yes‡ |
| 0.02 | 97.79 | 98.53 | 1.06 | 0.41 | +163 | No* |
| 0.03 | 96.58 | 97.05 | 1.49 | 0.82 | +82 | No* |
| 0.04 | **96.32** | 96.56 | 1.25 | 1.26 | −0.5 | Yes |
| 0.05 | **95.83** | 95.33 | 1.71 | 1.70 | +0.6 | Yes |
| 0.06 | **95.10** | 94.35 | 2.60 | 2.06 | +26 | Yes |
| 0.07 | **95.83** | 93.61 | 2.61 | 2.35 | +11 | Yes |
| 0.08 | **96.08** | 92.14 | 3.08 | 2.92 | +5 | Yes |
| 0.09 | **93.87** | 91.39 | 4.38 | 3.61 | +21 | Yes |
| 0.10 | **92.65** | 90.17 | 4.70 | 4.05 | +16 | Yes |

*DCF misses coverage target by less than 0.5 pp.

‡Degenerate retention: at $\alpha = 0.01$ the calibration set is too small to support the quantile, so the conservative fallback retains zero claims (Limitations).

## D.5. Complete Results Tables

# E. Ablation Study Details

## E.1. Single-Feature Baseline Configuration

For each single-feature baseline, we use CF with $\beta_{\mathrm{mix}}$ optimized per-$\alpha$ via grid search over $\{0.0, 0.1, \ldots, 1.0\}$. Table 11 shows the selected values.

*Table 11.* Optimized $\beta_{\text{mix}}$ values for MATH single-feature baselines.

| $\alpha$ | NX Reachability | Claim Index | Frequency Score |
|------|------|------|------|
| 0.03 | 0.7 | 0.9 | 0.0 |
| 0.04 | 0.9 | 1.0 | 0.0 |
| 0.05 | 1.0 | 0.7 | 1.0 |
| 0.06 | 0.3 | 1.0 | 0.4 |
| 0.07 | 1.0 | 1.0 | 0.3 |
| 0.08 | 0.2 | 0.9 | 0.1 |
| 0.09 | 0.5 | 1.0 | 1.0 |
| 0.10 | 0.6 | 1.0 | 0.9 |

*Table 12.* Optimized $\beta_{\text{mix}}$ values for FELM single-feature baselines.

| $\alpha$ | NX Reachability | Claim Index | Frequency Score |
|------|------|------|------|
| 0.01 | 0.0 | 0.5 | 0.8 |
| 0.02 | 0.0 | 0.2 | 0.6 |
| 0.03 | 0.2 | 0.4 | 0.5 |
| 0.04 | 0.9 | 1.0 | 0.2 |
| 0.05 | 0.7 | 0.9 | 0.4 |
| 0.06 | 0.5 | 1.0 | 0.3 |
| 0.07 | 0.0 | 1.0 | 0.4 |
| 0.08 | 0.3 | 1.0 | 0.1 |
| 0.09 | 0.0 | 0.8 | 0.1 |
| 0.10 | 0.0 | 1.0 | 0.0 |

## E.2. Complete Numerical Results

Tables 13 and 14 provide complete retention and coverage results for MATH and FELM respectively.

*Table 13.* MATH ablation: DCF vs. single-feature baselines with optimized $\beta_{\text{mix}}$.

| $\alpha$ | DCF (Ours) | | NX Reachability | | Claim Index | | Frequency | |
|------|------|------|------|------|------|------|------|------|
| | Cov | Ret | Cov | Ret | Cov | Ret | Cov | Ret |
| 0.03 | 0.97 | **1.76** | 0.98 | 1.60 | 0.98 | 0.93 | 0.98 | 0.47 |
| 0.04 | 0.96 | **2.22** | 0.97 | 2.04 | 0.97 | 1.19 | 0.97 | 0.84 |
| 0.05 | 0.96 | **2.31** | 0.96 | 2.16 | 0.96 | 1.41 | 0.97 | 1.04 |
| 0.06 | 0.94 | **3.56** | 0.94 | 2.87 | 0.95 | 1.86 | 0.95 | 1.49 |
| 0.07 | 0.93 | **3.63** | 0.94 | 2.69 | 0.93 | 2.54 | 0.94 | 1.81 |
| 0.08 | 0.94 | **3.69** | 0.93 | 3.26 | 0.93 | 2.67 | 0.93 | 1.94 |
| 0.09 | 0.91 | **5.57** | 0.91 | 3.72 | 0.93 | 3.37 | 0.93 | 2.20 |
| 0.10 | 0.90 | **5.99** | 0.91 | 4.18 | 0.91 | 3.62 | 0.92 | 2.21 |

## Key Observations (MATH).

- DCF outperforms all single-feature baselines at every $\alpha$ value
- Improvement over NX Reachability (best single feature): 7–50%
- Largest gains at strict coverage: +50% at $\alpha = 0.09$
- All methods maintain target coverage within acceptable margins

## Key Observations (FELM).

- DCF outperforms all single-feature baselines at low $\alpha$ (0.01–0.05) and high $\alpha$ (0.09–0.10)
- Frequency Score baseline wins at mid-range $\alpha$ (0.06–0.08), reflecting FELM's simpler graph structure
- DCF improvement over best baseline: up to +41% at $\alpha = 0.01$

*Table 14.* FELM ablation: DCF vs. single-feature baselines with optimized $\beta_{\mathrm{mix}}$. Retention is mean claims per problem.

| | DCF (Ours) | | NX Reachability | | Claim Index | | Frequency | |
|---|---|---|---|---|---|---|---|---|
| $\alpha$ | Cov | Ret | Cov | Ret | Cov | Ret | Cov | Ret |
| 0.01 | 0.99 | **0.72** | 0.99 | 0.17 | 0.99 | 0.24 | 0.99 | 0.51 |
| 0.02 | 0.98 | **1.43** | 0.98 | 0.31 | 0.98 | 0.33 | 0.98 | 0.87 |
| 0.03 | 0.97 | **1.46** | 0.97 | 0.50 | 0.97 | 0.44 | 0.97 | 1.27 |
| 0.04 | 0.96 | **1.68** | 0.96 | 0.58 | 0.96 | 0.61 | 0.96 | 1.55 |
| 0.05 | 0.95 | **1.85** | 0.95 | 0.73 | 0.95 | 0.72 | 0.95 | 1.71 |
| 0.06 | 0.95 | 1.92 | 0.94 | 0.84 | 0.94 | 0.86 | 0.94 | **2.04** |
| 0.07 | 0.94 | 1.98 | 0.93 | 0.96 | 0.93 | 1.05 | 0.93 | **2.27** |
| 0.08 | 0.94 | 2.11 | 0.92 | 1.05 | 0.92 | 1.26 | 0.92 | **2.56** |
| 0.09 | 0.91 | **2.76** | 0.91 | 1.22 | 0.91 | 1.31 | 0.91 | 2.72 |
| 0.10 | 0.91 | **2.85** | 0.90 | 1.29 | 0.90 | 1.50 | 0.90 | 2.80 |

- Frequency Score is a much stronger single-feature baseline on FELM than on MATH

### E.3. Scale Robustness and Scorer Capacity

This appendix presents the MATH-407 scale-robustness evaluation and the scorer-capacity case study at $\alpha = 0.09$.

#### E.3.1. SCALE ROBUSTNESS ON MATH-407

To validate that our findings generalize beyond the MATH-202 evaluation, we re-evaluated DCF and all baselines on a $2\times$ expansion of MATH (407 manually annotated problems, denoted MATH-407) using the identical 20-fold cross-validation protocol. Figure 8 shows retention and coverage across $\alpha \in [0.01, 0.10]$. DCF (Logistic) outperforms the optimized frequency-CF baseline at 8 of 9 non-degenerate $\alpha$ values, with gains ranging from $+3\%$ to $+121\%$ (mean $+31\%$), confirming that the qualitative results scale. The coverage curve continues to closely track the $1 - \alpha$ target, indicating that the conformal guarantee is preserved at the larger scale. Notably, NX Reachability—the strongest hand-crafted single-feature CF variant from Section 4.4—becomes more competitive with DCF (Logistic) at moderate $\alpha$ on the larger calibration set, since simple discriminative features benefit from increased calibration support. This motivates the scorer-capacity study below, which examines whether richer scorer architectures can recover the gap. Per-$\alpha$ numerical results appear in Table 10.

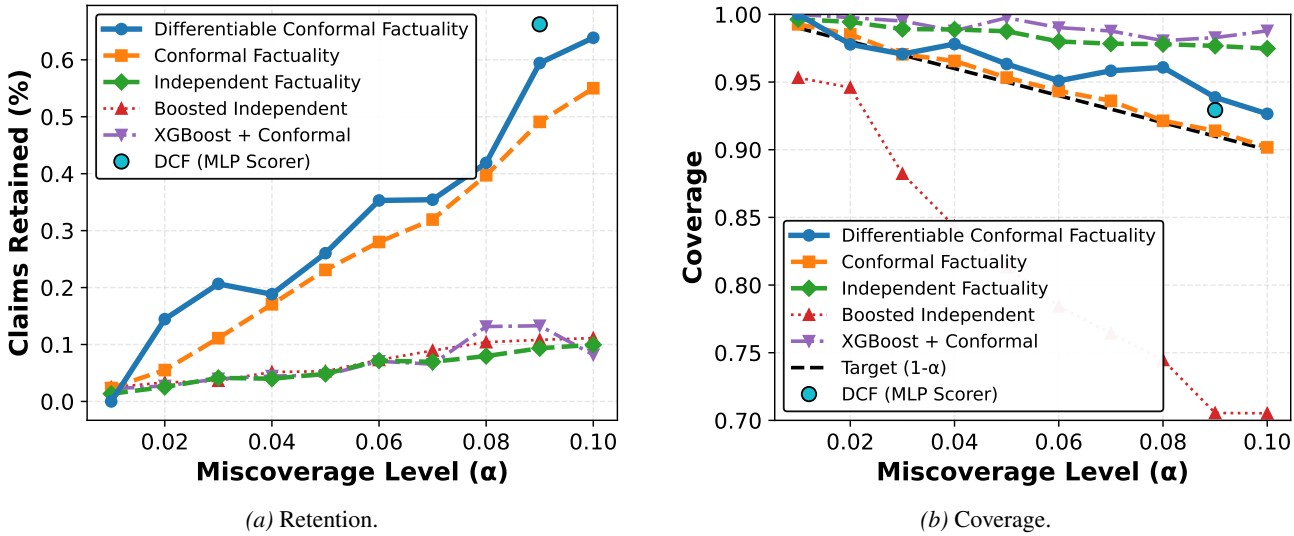

*(a)* Retention.  *(b)* Coverage.

*Figure 8.* DCF vs. baselines on MATH-407.

E.3.2. SCORER CAPACITY CASE STUDY AT $\alpha = 0.09$

A natural question is whether DCF's linear scorer leaves expressive signal on the table: do richer scorer architectures recover the gap to the strongest hand-crafted CF variants? We test this at $\alpha = 0.09$, where all methods retain meaningful coverage margin, by replacing the linear scorer with a shallow MLP (1 hidden layer, $h{=}32$, dropout 0.2) and re-tuning $\gamma, \lambda$ via grid search. Table 15 reports 20-fold paired comparisons.

*Table 15.* Scorer capacity at $\alpha = 0.09$ on MATH-407 (20-fold CV, paired). $\Delta$ shows percentage retention improvement over frequency-CF. Significance from paired $t$-test against frequency-CF; $\beta_{\mathrm{mix}}$ optimized per single-feature CF baseline (Appendix D.5).

| Method | Coverage | Retention | $\Delta$ |
|---|---|---|---|
| CF (Frequency) | 0.914 | 3.618 | — |
| CF (NX Reachability) | 0.912 | 4.332 | +20% |
| DCF (Logistic) | **0.939** | 4.375 | +21%[*] |
| **DCF (MLP, $h{=}32$)** | **0.929** | **4.876** | **+35%**[***] |

[*]$p < 0.05$, [**]$p < 0.01$, [***]$p < 0.001$ (paired $t$-test).

DCF with the MLP scorer retains 4.876 claims/problem, outperforming NX Reachability by $+12.6\%$ (paired $t$-test $p = 0.001$, Cohen's $d = 0.87$, MLP wins 16/20 folds) and the linear DCF scorer by $+11.5\%$ ($p = 0.002$, $d = 0.79$, 16/3 wins with 1 tie). This demonstrates that DCF readily benefits from richer scorer architectures when calibration data is sufficient; we retain the linear scorer as the main result for interpretability, while noting that scorer capacity is a straightforward axis of further improvement.

E.3.3. HIDDEN-DIMENSION SWEEP

To validate the architectural choice in the scorer-capacity case study (Section E.3.2), we swept the MLP hidden dimension while holding all other DCF hyperparameters fixed at the best MLP configuration ($\gamma = 1.5$, $\lambda = 2.30$, dropout$= 0.2$). Table 16 reports 20-fold paired comparisons against the Logistic scorer (using its own DCF-optimal $\gamma = 4.0$, $\lambda = 2.30$).

*Table 16.* Hidden-dimension sweep at $\alpha = 0.09$ on MATH-407 (20-fold CV, paired). $\Delta$ shows mean retention difference versus Logistic. Significance from paired $t$-test against Logistic; ns = not significant.

| Scorer | Params | Coverage | Retention | $\Delta$ vs. Logistic | $p$ |
|---|---|---|---|---|---|
| Logistic (linear) | 31 | 0.939 | 4.375 | — | — |
| MLP $h{=}8$, $d{=}0.2$ | 257 | 0.927 | 4.168 | $-0.207$ | ns |
| MLP $h{=}16$, $d{=}0.2$ | 513 | 0.934 | 4.701 | $+0.325$ | ns |
| **MLP $h{=}32$, $d{=}0.2$** | 1025 | 0.929 | **4.876** | **+0.501** | ** |
| MLP $h{=}64$, $d{=}0.2$ | 2049 | 0.934 | 4.409 | $+0.033$ | ns |

[*]$p < 0.05$, [**]$p < 0.01$, [***]$p < 0.001$ (paired $t$-test).

The sweep reveals a clear capacity threshold around $h{=}32$: $h{=}8$ underfits (no significant gain over Logistic), $h{=}16$ improves modestly but not significantly, $h{=}32$ achieves a statistically significant retention gain ($+0.501$, $p < 0.01$), and $h{=}64$ regresses back toward Logistic-level retention—consistent with diminishing returns or mild overfitting at higher capacity. All configurations satisfy the 0.91 coverage target. We adopt $h{=}32$ for the case study based on these results.

**E.4. Hyperparameter Sensitivity**

To characterize the practical sensitivity of DCF to its temperature parameters, we sweep each hyperparameter independently at $\alpha = 0.06$ on MATH (20-fold CV), holding all other parameters at their per-$\alpha$ optimized values. Table 17 reports retention and coverage averaged across eight $\alpha$ levels.

**Caveat on $\tau_s$.** This sensitivity sweep explicitly invokes the exponential violation form (Eq. 4); however, our main optimization results use the min-max variant. The zero sensitivity observed for $\tau_s$ here reflects the exponential form's behavior at well-trained DCF configurations: $\log Q_\tau$ values for violating thresholds become highly negative, so $\exp(\log Q_\tau/\tau_s) \approx 0$ for any reasonable $\tau_s$, producing $V_\tau \approx 1$ regardless of the precise scale. Either way—exponential or min-max—practical retention is insensitive to $\tau_s$.

**Key findings.**

*Table 17.* Hyperparameter sensitivity on MATH (202 examples, 20-fold CV). Each parameter is swept independently while all others are held at their per-$\alpha$ optimized values. Values show mean $\pm$ std averaged across all 8 $\alpha$ levels.

| Parameter | Value | Retention | Coverage |
|---|---|---|---|
| $T_p$ | 0.01 | $3.30 \pm 1.70$ | $0.937 \pm 0.030$ |
| $T_p$ | 0.1 | $3.66 \pm 1.36$ | $0.934 \pm 0.021$ |
| $T_p$ | 0.5 | $3.35 \pm 1.36$ | $0.924 \pm 0.028$ |
| $T_p$ | 1 | $3.28 \pm 1.41$ | $0.932 \pm 0.031$ |
| $T_p$ | 2 | $3.00 \pm 1.70$ | $0.923 \pm 0.041$ |
| $\tau_s$ | 0.1 | $3.59 \pm 1.44$ | $0.937 \pm 0.023$ |
| $\tau_s$ | 0.4 | $3.59 \pm 1.44$ | $0.937 \pm 0.023$ |
| $\tau_s$ | 1 | $3.59 \pm 1.44$ | $0.937 \pm 0.023$ |
| $\tau_s$ | 2 | $3.59 \pm 1.44$ | $0.937 \pm 0.023$ |
| $\tau_s$ | 5 | $3.59 \pm 1.44$ | $0.937 \pm 0.023$ |
| $\beta$ | 0.5 | $3.21 \pm 1.63$ | $0.922 \pm 0.027$ |
| $\beta$ | 1 | $3.59 \pm 1.44$ | $0.937 \pm 0.023$ |
| $\beta$ | $\{5, 10, 20\}$ | degenerate (all claims rejected) | |
| $\lambda$ | 0.5 | $3.03 \pm 1.72$ | $0.902 \pm 0.037$ |
| $\lambda$ | 0.8 | $2.98 \pm 1.71$ | $0.908 \pm 0.037$ |
| $\lambda$ | 1 | $2.98 \pm 1.70$ | $0.912 \pm 0.037$ |
| $\lambda$ | 1.2 | $3.11 \pm 1.61$ | $0.921 \pm 0.035$ |
| $\lambda$ | 1.35 | $3.31 \pm 1.47$ | $0.925 \pm 0.034$ |
| $\lambda$ | 1.5 | $3.51 \pm 1.43$ | $0.917 \pm 0.027$ |
| $\lambda$ | 1.7 | $3.29 \pm 1.54$ | $0.935 \pm 0.019$ |
| $\lambda$ | 1.9 | $3.00 \pm 1.75$ | $0.907 \pm 0.027$ |
| $\lambda$ | 2.5 | $2.95 \pm 1.71$ | $0.899 \pm 0.037$ |
| $\gamma$ | 0.5 | $3.43 \pm 1.44$ | $0.929 \pm 0.023$ |
| $\gamma$ | 1 | $3.47 \pm 1.45$ | $0.929 \pm 0.026$ |
| $\gamma$ | 1.5 | $3.49 \pm 1.40$ | $0.928 \pm 0.022$ |
| $\gamma$ | 2 | $3.52 \pm 1.40$ | $0.935 \pm 0.024$ |
| $\gamma$ | 4 | $3.49 \pm 1.40$ | $0.935 \pm 0.023$ |
| $\gamma$ | 5 | $3.53 \pm 1.44$ | $0.937 \pm 0.023$ |
| $\gamma$ | 6 | $3.55 \pm 1.38$ | $0.937 \pm 0.022$ |
| $\gamma$ | 8 | $3.56 \pm 1.44$ | $0.938 \pm 0.023$ |

- $T_p$: moderate sensitivity (best at 0.1; robust across 0.01–1.0).
- $\gamma$: very low sensitivity (range $\approx 0.13$ across $\gamma \in [0.5, 8]$).
- $\lambda$: moderate (peaks around 1.5).
- $\beta$: requires care; $\beta \in \{0.5, 1.0\}$ work, $\beta \geq 5$ causes coverage collapse.
- $\tau_s$: no measurable effect (see caveat above).

No scheduling is needed; fixed values within the working ranges above are sufficient.

### E.5. Hypothesized Mechanism for FELM's Residual Gains

The MI analysis in Section 4.3.1 explains the cross-dataset magnitude difference, but leaves a residual puzzle on FELM: DCF still wins at some $\alpha$ values despite orthogonal MI being near zero. The mechanism we hypothesize is *objective alignment at the operating $\alpha$*. While the MI analysis characterizes globally-averaged discriminative information, conformal retention depends on the score's behavior near a single threshold—the $(1 - \alpha)$-quantile—and DCF is trained with a differentiable surrogate that concentrates gradient signal on this region rather than producing a globally well-ordered ranking. The learned score can discriminate near thresholds even when residual signal is small. In contrast, FELM gains are inconsistent across $\alpha$ because the magnitude of the improvement depends on the score-distribution shape near each specific threshold.

## F. Case Study Details

### F.1. Methodology

The case study in Section 4.4.2 is generated using the following methodology:

**Example Selection.** We choose a representative example from the MATH dataset at $\alpha = 0.06$: **Example 186, Claim 8**, a true claim with frequency-score = 0 that the learned model correctly retains.

**Evaluation Protocol.** We use 20-fold cross-validation matching the main experimental setup:

1. For each fold, train the learned logistic claim scorer on the training set
2. Compute calibration quantiles for both learned and baseline methods
3. Generate prediction sets for the test examples using both methods
4. Record which claims are retained in each fold's prediction set

**Baseline Configuration.** The frequency-based CF baseline uses $\beta_{\text{mix}}$ optimized per-$\alpha$ via grid search over $\{0.0, 0.1, \ldots, 1.0\}$. At $\alpha = 0.06$, the optimal value is $\beta_{\text{mix}} = 0.4$ (see Table 11).

**Majority Voting.** For visualization, we use majority voting across folds: a claim is considered "retained" if it appears in the prediction set in more than 50% of folds ($> 10$ out of 20 folds). This provides stable results despite fold-to-fold variation.

**Feature Contributions.** Model weights are averaged across the 20 folds to compute stable feature contributions. The contribution of each feature is computed as weight $\times$ value, and the total learned score is the sum of all contributions plus bias.

## F.2. Feature Contribution Tables

Tables 18 and **??** show feature contributions for the case study. Only features with non-zero values are shown; the remaining features (primarily domain indicators not applicable to the specific problem) have value 0 and contribute nothing to the score.

*Table 18.* Case Study (Example 186, Claim 8): Feature contributions for valid claim with zero frequency.

| Feature | Value | Weight | Contribution |
|---|---|---|---|
| nx_reachability | 3.00 | 0.272 | +0.815 |
| claim_index | 8.00 | 0.098 | +0.780 |
| nx_in_degree | 2.00 | 0.135 | +0.269 |
| quadratic_equations | 1.00 | 0.229 | +0.229 |
| nx_out_degree | 1.00 | 0.176 | +0.176 |
| problem_relevance | 1.00 | 0.144 | +0.144 |
| coherent_to_ancestors | 1.00 | 0.110 | +0.110 |
| nx_betweenness | 0.22 | 0.292 | +0.064 |
| uses_problem_data | 0.50 | 0.104 | +0.052 |
| frequency-score | 0.00 | 0.021 | +0.000 |
| **Total Learned Score** | | | **2.66** |
| **Baseline Score (freq-score)** | | | 0.00 |
| **Prediction Set Retention** | | Learned: 9/12, Baseline: 0/12 | |

## F.3. Analysis

**Retaining Valid Claims with Zero Frequency.** This case study appears in the main text (Section 4.4.2). Graph connectivity features (nx_reachability +0.82, nx_in_degree +0.27) and claim position (claim_index +0.78) compensate for zero frequency-score, enabling retention of a valid intermediate reasoning step that the baseline would reject entirely.

## F.4. DCF-Guided Reprompting: A Detect-and-Repair Workflow

A natural concern is that DCF only *filters* hallucinated claims; it does not *fix* them. We demonstrate that DCF's rejection signal can drive a downstream repair workflow that produces fully coherent solutions. Consider MATH example 79 at

$\alpha = 0.04$, where all 10 claims of the original solution have frequency-score $= 5.0$ (the LLM consistently regenerates them), but only 2 are coherently factual—intermediate algebraic steps are skipped, breaking the dependency chain. The frequency-CF baseline retains all 10 claims (precision $= 0.20$) while DCF retains 1 (the verified root; precision $= 1.00$).

Using DCF's per-claim rejections, we reprompt the LLM with a structured request: for each rejected claim, identify the largest reasoning gap to its dependency parents and insert the minimum number of bridging intermediate steps. The reprompt explicitly states that flagged claims are likely arithmetically correct but compress too many algebraic manipulations into a single step. The LLM produces a stitched 12-claim solution with two bridging intermediates inserted, restoring full coherence (verified by reannotation). Crucially, the rejection signal is what enables the repair: without DCF's per-claim verdict, the LLM has no signal indicating which steps need expansion.

This demonstrates that DCF integrates naturally into agentic detect-and-repair workflows—identifying logical gaps in reasoning chains and feeding them back as repair targets. We view this as a more practical mode of deployment than filter-only use, especially in interactive settings.

### F.5. Methodology

To ensure robust feature importance estimates, we aggregate SHAP values across multiple coverage levels and training splits. We analyze models trained at $\alpha \in \{0.03, 0.04, 0.05, 0.06, 0.07, 0.08, 0.09, 0.10\}$. For each $\alpha$, we train 5 independent models using different train/validation splits (80/20), yielding 40 total models (8 $\alpha$ values $\times$ 5 splits).

**SHAP Computation.**

- Method: Kernel SHAP (model-agnostic explainer)
- Background data: 100 samples randomly selected from all claims
- Evaluation samples: Maximum 500 claims per model
- For LogisticClaimScorer, we extract linear weights $w$ and bias $b$ to create prediction function $f(x) = xw + b$

**Aggregation Procedure.**

1. Collect SHAP values from all 40 models (each contributing up to 500 claim samples)
2. Pool all SHAP values per feature across models ($\sim$20,000 values per feature)
3. Compute mean absolute SHAP value: importance$_i$ = mean($|\text{SHAP}_i|$)
4. Compute standard deviation to quantify variance across models
5. Rank features by mean importance

### F.6. Results

Table 19 shows the top 10 features by mean |SHAP| value.

*Table 19.* Top 10 features by SHAP importance aggregated across 40 models.

| Rank | Feature | Mean |SHAP| | Std |
|------|---------|-------------|-----|
| 1 | nx_reachability | 0.602 | 0.428 |
| 2 | claim_index | 0.445 | 0.303 |
| 3 | frequency-score | 0.180 | 0.126 |
| 4 | inference_gap_size | 0.124 | 0.248 |
| 5 | nx_out_degree | 0.060 | 0.081 |
| 6 | nx_in_degree | 0.038 | 0.056 |
| 7 | nx_is_source | 0.034 | 0.035 |
| 8 | problem_relevance | 0.034 | 0.032 |
| 9 | uses_problem_data | 0.034 | 0.041 |
| 10 | quadratic_equations | 0.022 | 0.020 |

**Key Findings.**

- **Graph structure dominates**: nx_reachability (0.602) contributes 3.3$\times$ more than frequency-score (0.180)

- **Position matters**: `claim_index` (0.445) contributes 2.5× more than frequency
- **Coherence features rank highly**: `inference_gap_size` (0.124) captures missing reasoning steps
- **Mathematical content has lower impact**: Domain indicators individually contribute less, but collectively capture topic-specific patterns

Figure 9 shows the complete SHAP beeswarm plot with all 30 features.

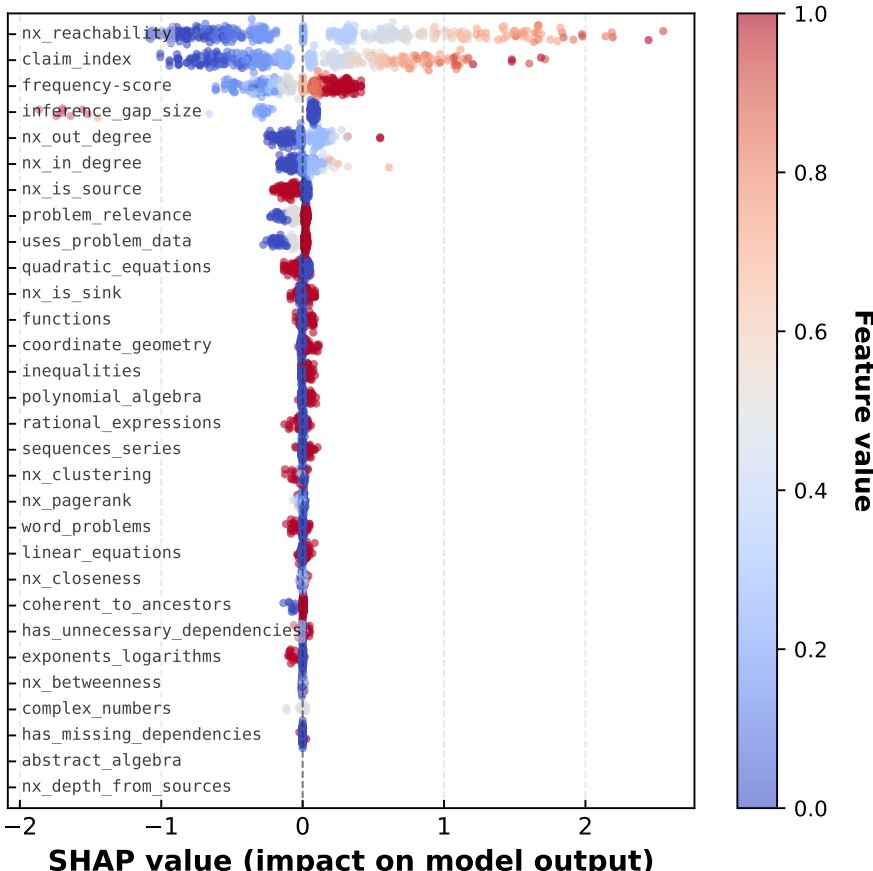

*Figure 9.* MATH SHAP beeswarm plot across 40 models. Each dot represents one claim; color indicates normalized feature value (blue=low, red=high). Graph structure features dominate over frequency-based scoring.

### F.7. FELM SHAP Analysis

For FELM, we perform separate SHAP analyses for the two feature configurations used at different $\alpha$ levels: 7 features for $\alpha \in [0.01, 0.08]$ and 20 features for $\alpha \in [0.09, 0.10]$.

**7-Feature Models ($\alpha \leq 0.08$).** The 7-feature configuration uses: `frequency-score`, `coherent_to_ancestors`, `inference_gap_size`, `has_missing_dependencies`, `nx_pagerank`, `nx_reachability`, and `nx_out_degree`. Figure 10a shows that `frequency-score` dominates (0.995), with `nx_reachability` (0.154) and `inference_gap_size` (0.093) as secondary contributors.

**20-Feature Models ($\alpha \geq 0.09$).** At higher $\alpha$ values, we use the full 20-feature set including additional graph metrics and coherence features. Figure 10b shows that `frequency-score` remains dominant (0.941), but `claim_index` (0.209) and `nx_reachability` (0.149) gain importance with the expanded feature set.

**MATH vs. FELM Comparison.** The key difference between datasets is the relative importance of `frequency-score`: it ranks 3rd on MATH (0.180) but 1st on FELM (0.941–0.995). This reflects FELM's simpler reasoning chains (4.0 vs.

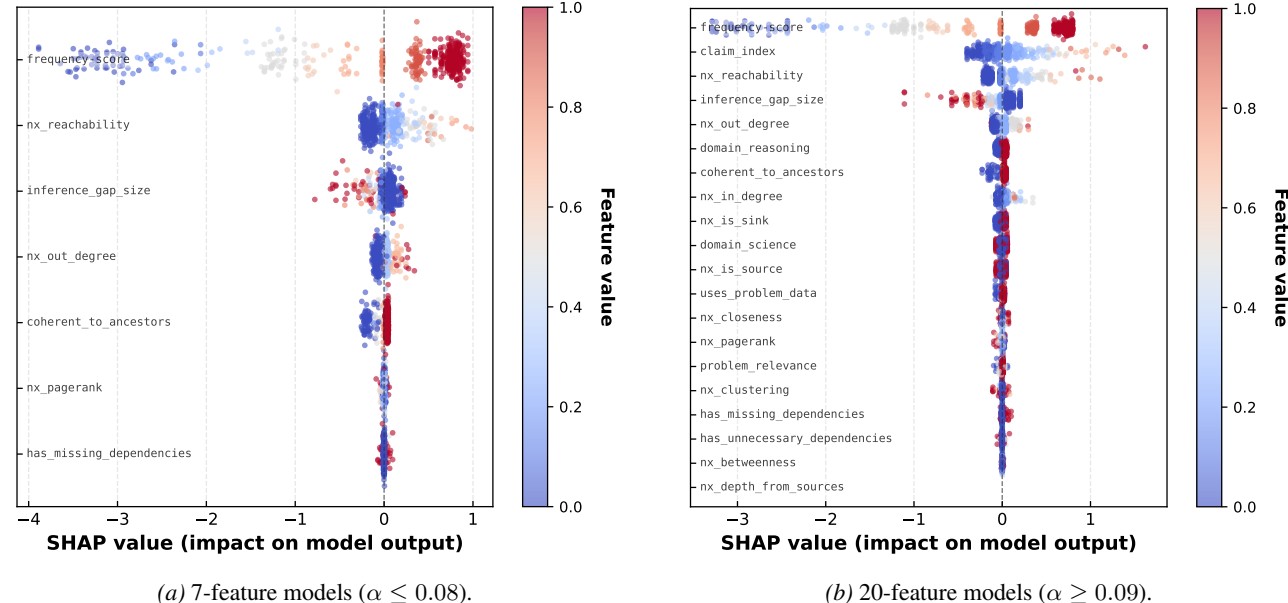

*(a)* 7-feature models ($\alpha \leq 0.08$).          *(b)* 20-feature models ($\alpha \geq 0.09$).

*Figure 10.* FELM SHAP beeswarm plots. Left: 7-feature configuration used at stricter coverage levels. Right: 20-feature configuration used at relaxed coverage levels. `frequency-score` dominates in both, reflecting FELM's simpler reasoning chains where self-consistency is more discriminative.

7.3 claims per problem), where self-consistency frequency is more discriminative. On MATH, graph structure features (`nx_reachability`, `claim_index`) provide stronger signal for navigating complex dependency graphs.

## G. Score Distribution Analysis

To understand how DCF differentiates true from false claims, we analyze score distributions using pooled z-score normalization.

### G.1. Methodology

For each $\alpha$ value, we:

1. Collect scores from all 20 CV folds for both learned and baseline models
2. Apply z-score normalization pooled across both methods for comparability
3. Compute separation as: sep $= \mu_{\text{true}} - \mu_{\text{false}}$
4. Compute Cohen's $d$ effect size and distribution overlap

### G.2. Results

Figure 11 shows the 2×2 comparison at $\alpha = 0.05$.

Table 20 shows separation metrics across all $\alpha$ values.

**Interpretation.** Separation varies substantially—and can even be negative—because DCF optimizes for coverage-constrained retention, not distributional separation. At $\alpha = 0.09$, DCF achieves +103% higher retention despite negative separation, demonstrating that conformal prediction succeeds through calibrated threshold placement rather than global score discrimination. The quantile-based approach requires only that the threshold correctly partitions claims at the decision boundary, not that true and false claims be globally separated.

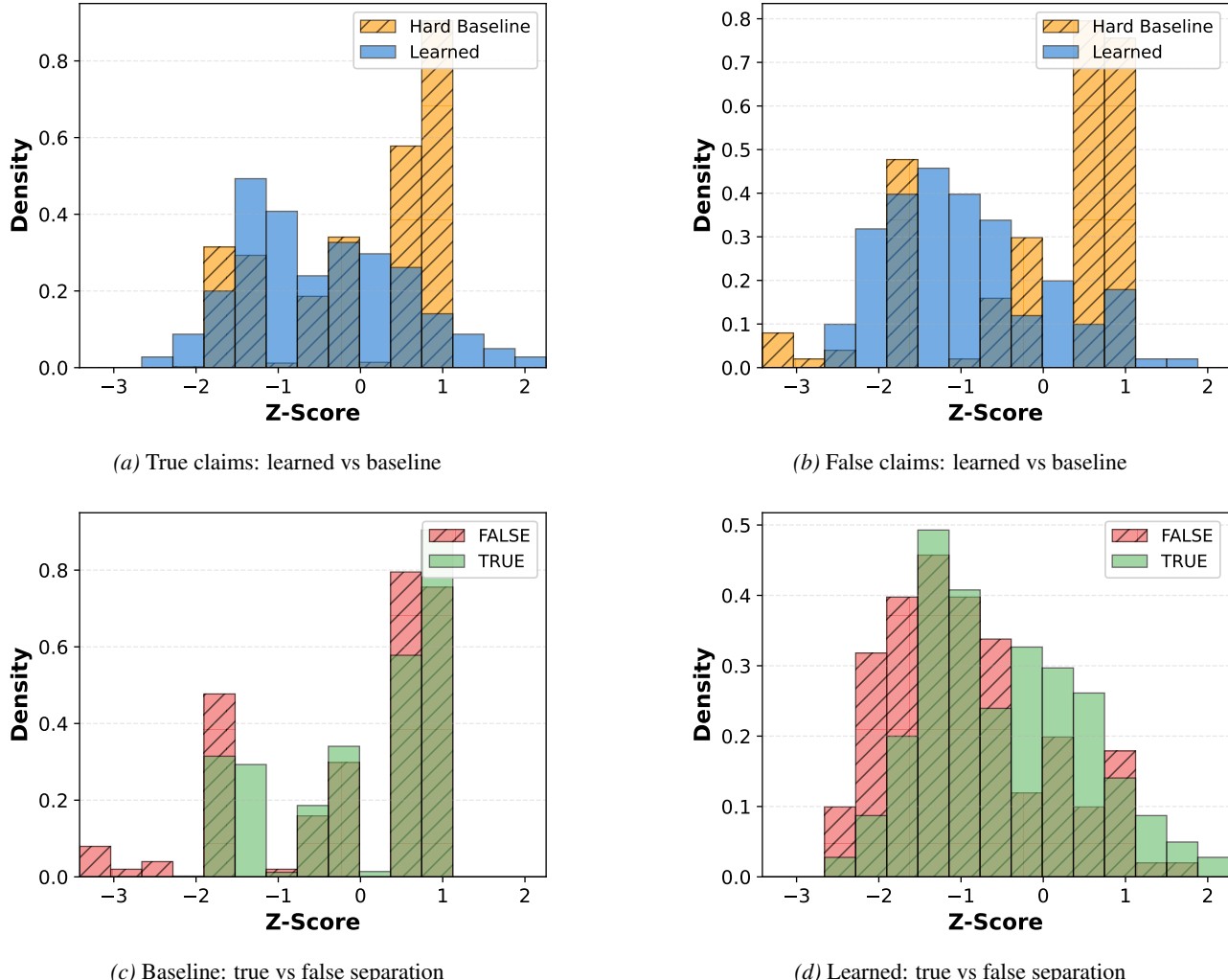

*(a)* True claims: learned vs baseline

*(b)* False claims: learned vs baseline

*(c)* Baseline: true vs false separation

*(d)* Learned: true vs false separation

*Figure 11.* Score distributions at $\alpha = 0.05$. Top row: method comparison for true/false claims. Bottom row: within-method discrimination. The learned model achieves separation 0.450 vs. baseline's 0.141 (3.2×).

*Table 20.* Score separation metrics across $\alpha$ values. DCF optimizes retention under coverage constraints, not separation directly, explaining the variability.

| $\alpha$ | Separation | Ratio vs. Baseline | Cohen's $d$ | Overlap |
|------|------|------|------|------|
| 0.03 | 0.170 | 1.21× | 0.298 | 0.926 |
| 0.04 | 0.040 | 0.28× | 0.046 | 0.962 |
| 0.05 | 0.450 | 3.18× | 0.468 | 0.939 |
| 0.06 | 0.056 | 0.40× | 0.057 | 0.954 |
| 0.07 | 0.535 | 3.80× | 0.514 | 0.882 |
| 0.08 | 0.434 | 3.05× | 0.574 | 0.927 |
| 0.09 | −0.184 | – | −0.192 | 0.933 |
| 0.10 | 0.111 | 0.79× | 0.124 | 0.945 |

