# OpenReview forum: "Differentiable Conformal Training for LLM Reasoning Factuality"
_ICML.cc/2026/Conference — ICML 2026 regular_

### Official Review · Reviewer_GM3f · 2026-03-11

**Soundness:** 3
**Presentation:** 3
**Significance:** 3
**Originality:** 4
**Overall Recommendation:** 4
**Confidence:** 4

**Summary:**

The paper addresses the challenge of hallucination in Large Language Models (LLMs) by improving upon Conformal Prediction (CP) methods that provide statistical guarantees on factuality. While prior work like Coherent Factuality (CF) accounts for logical dependencies in multi-step reasoning using dependency graphs, it relies on hand-crafted scoring functions that are overly conservative, often filtering out up to 60% of true claims to maintain reliability.

The authors propose Differentiable Coherent Factuality (DCF), a framework that introduces differentiable relaxations for the discrete operations in CF—specifically threshold filtering, ancestor coherence, and argmax selection. This allows for the end-to-end gradient-based optimization of risk scorers while theoretically and empirically preserving the original algorithm's coverage guarantees. Experiments on MATH and FELM datasets show that DCF significantly improves claim retention (up to 141% on MATH) compared to non-learned baselines.

**Compliance With Llm Reviewing Policy:**

Affirmed.

**Key Questions For Authors:**

- How does the requirement for manual annotation of dependency graphs (ADGs) affect the scalability of DCF to new domains? Can LLMs reliably generate these graphs in a zero-shot manner without human verification?
- Why does DCF's performance fluctuate on the FELM dataset at mid-range $\alpha$ values compared to the more consistent improvements seen on the MATH dataset?
- The current objective maximizes the number of true claims retained. Have you explored alternative loss functions that prioritize "pivotal" claims in a reasoning chain over simple atomic count?
- Could you provide more details on the sensitivity of the results to the temperature parameters ($T_p, \tau_s, \beta$)? Is there a recommended scheduling strategy during training to ensure convergence?

**Limitations:**

- As with all conformal methods, the guarantees are marginal (holding on average) rather than per-prediction, which might be insufficient for some critical safety applications.
- DCF can only retain claims that the underlying LLM actually generates; it does not "fix" hallucinations but merely filters them out.
- The reliance on a held-out calibration set means that performance is bounded by the quality and size of that set, with small sets leading to the aforementioned quantile instability.
- The gains are most significant when frequency is a poor discriminator of truth; in tasks where self-consistency is highly reliable, the overhead of DCF may not be justified.

**Strengths And Weaknesses:**

Strengths

- The authors provide proofs (Theorems 3.1 and 3.2) showing that their differentiable relaxations recover the original discrete CF algorithm in the limit, ensuring that optimization translates to real-world performance without sacrificing statistical guarantees.
- DCF demonstrates substantial gains in utility (claim retention) over state-of-the-art methods like CF and independent claim filtering, especially at high reliability levels ($\alpha < 0.1$).
- The use of a logistic regression-based scorer over 30 claim-level features allows for SHAP analysis, revealing that graph-structured signals (like reachability) are often more predictive than standard self-consistency frequency scores.
- The work successfully bridges the gap between formal statistical guarantees and the practical usability of LLM outputs by reducing the "conservativeness" of the filtering process.

Weaknesses

- The method struggles at extremely strict error rates (e.g., $\alpha \le 0.02$). In these cases, the small number of calibration examples with errors makes the quantile estimator unstable, often leading the system to reject all claims.
- On datasets with simpler reasoning chains like FELM, the advantage of DCF over frequency-based baselines is less pronounced, and the baselines occasionally outperform DCF at mid-range $\alpha$ values.
- The MATH dataset evaluation is limited to 202 problems with manual annotations, which may raise questions about the robustness of the findings across broader or more diverse reasoning tasks.
- The joint relaxation of tightly coupled discrete operations requires careful temperature parameter tuning to encode the correct algorithmic ordering, which may complicate implementation.

---

> ### Author Rebuttal · Authors · 2026-03-29
>
> **Rebuttal to Reviewer GM3f**
>
> We address each concern with new experiments and cross-references to our other responses.
>
> **W1 (α-collapse), W2 (FELM gains), W3 (dataset size):** These concerns are related. We understand why our presentation of α-collapse may have suggested a limitation; it is a deliberate safeguard, which we detail in our response to Reviewer zMxT (full analysis, minimum n_cal table, MATH-407 results where DCF beats optimized CF at every non-degenerate α, +2% to +115%). FELM's smaller gains are explained by a mutual information analysis in our response to Reviewer Azq4: on FELM, structural features carry 0% orthogonal information beyond frequency, while on MATH they carry 636% more. DCF's advantage scales with reasoning complexity.
>
> **W4: Temperature Parameter Tuning**
>
> We see why Theorem 3.1's temperature ordering discussion may suggest practical tuning difficulty. To clarify: this constraint arises only in the limit proof; in practice it is not significant. We will clarify in the revision. To prove this, we sweep on MATH (202, 20-fold CV) at α = 0.06; sensitivity measured in claims retained:
>
> | Parameter | Sensitivity | Key Finding |
> |---|---|---|
> | τ_s | **None** | Zero effect across 0.1–5.0 |
> | γ | **Very low** | Range 0.13; slight monotonic increase |
> | T_p | **Moderate** | Range 0.66; best at 0.1, robust across 0.01–1.0 |
> | λ | **Moderate** | Range 0.56; peaks around 1.5 |
> | β | **Sensitive** | β ∈ {0.5, 1.0} work well; β ≥ 5 causes collapse |
>
> Only β requires care (should remain ≤1). No scheduling needed; fixed values work.
>
> **Key Questions**
>
> **Q1: Scalability of ADG Annotations**
>
> Three approaches to scalability: (1) In safety-critical domains, annotation is justified and only needed once; the trained scorer then deploys on unlabeled IID data via standard CP. (2) Semi-synthetic annotation where LLMs generate candidate ADGs with lightweight human verification, since dependency structure is more amenable to automation than correctness labels. (3) Domain-specific tools (proof-checkers, code execution) can automate claim verification within the graph; the ADG structure naturally decomposes verification into modular subtasks.
>
> **Q2: FELM Mid-Range α Fluctuation**
>
> Two factors combine. First, FELM's frequency score already separates TRUE from FALSE claims well (mean gap=5.39 vs MATH's 0.10), leaving DCF a narrow margin. Second, DCF's learned scorer introduces 2.34× higher fold-to-fold coverage variance. The optimizer compensates by overshooting coverage at mid-range α by 0.9–1.6pp, consuming the retention margin (rank correlation between coverage excess and retention loss: ρ=−0.952, p<0.0001). At low α, both methods are forced conservative; at high α, the budget absorbs the variance. The path forward is reducing scorer variance (ensembling, regularization).
>
> **Why DCF still helps on FELM despite 0% orthogonal MI.** The learned scorer acts as a **denoiser**: frequency is estimated from ~5 LLM samples and is inherently noisy. The redundant features are correlated with frequency but measured independently; combining them reduces variance (ensemble averaging). Crucially, these features are extracted from the same samples, so denoising is free; simply running more frequency samples would scale linearly in LLM inference cost. Future work: designing features with low MI to frequency (model entropy, cross-model consistency) to provide genuinely orthogonal signal beyond denoising.
>
> **Q3: Alternative Loss Functions for Pivotal Claims**
>
> This is a great idea. Natural extension: weight claims by graph centrality (PageRank, reachability) to prioritize "hub" claims; we will test this in the final version.
>
> **Q4, L3:** Temperature sensitivity addressed in W4 above. Calibration set size addressed in our response to Reviewer zMxT (MATH-407 demonstrates more data improves stability).
>
> **Limitations**
>
> **L1: Marginal vs. Per-Prediction Guarantees**
>
> This limitation is inherent to all conformal methods. DCF is compatible with conditional coverage extensions (Gibbs & Candès, 2021; Cherian et al., 2024), which we plan to explore in future work.
>
> **L2: DCF Filters but Doesn't Fix Hallucinations**
>
> DCF's rejection signal enables a **detect-and-repair** workflow. We conducted a case study where the model produces 10 arithmetically correct claims (all with frequency=5.0), yet only 2/10 are coherently factual due to skipped intermediate steps. Frequency-based CF retains all 10 (precision=0.20); DCF retains 1 (precision=1.00). Using DCF's rejection signal, we reprompt to insert bridging intermediates at the identified gap, producing a fully coherent solution. We will include this case study in the final version.
>
> **L4: Overhead When Frequency Is a Good Discriminator**
>
> We agree. However, frequency is a poor discriminator precisely where CF matters most: complex multi-step reasoning with systematic, self-consistent errors. As LLM deployment moves toward agentic workflows, these conditions become more prevalent.

---

### Official Review · Reviewer_zMxT · 2026-03-12

**Soundness:** 4
**Presentation:** 3
**Significance:** 4
**Originality:** 4
**Overall Recommendation:** 5
**Confidence:** 3

**Summary:**

This paper introduces a differential surrogate for Coherent Factuality (CF) to support conformal prediction of hallucination control in multi-step reasoning by LLMs. The proposed Differentiable Coherent Factuality (DCF) replaces CF’s discrete thresholding, enforces ancestor-coherence and argmax selection with smooth surrogates and proves a differentiable gradient flow optimization converging to the original CF optimization and preserves the CF’s error guarantee. Experiments on MATH and FELM show substantial retention gains (up to 141%) at comparable coverage and illustrate DCF learns to combine complementary features beyond CF’s hand-crafted feature scoring.

**Compliance With Llm Reviewing Policy:**

Affirmed.

**Final Justification:**

Thank you for the rebuttal. The dataset size remains low despite the expansion. Overall, I believe this is a good paper worthy of acceptance though as I had already indicated in my original review, so I will not raise the score further.

**Key Questions For Authors:**

Please see the weaknesses.

**Limitations:**

Yes

**Strengths And Weaknesses:**

Strengths:
- The proposed method is well motivated by CF’s limited handcrafted frequency-based score and the need for ancestor-dependent algorithmic ordering and relaxation.
- The proposed method, DCF, introduces a joint relaxation of CF’s coupled discrete steps, enforcing ancestor coherence rather than per-claim discrete relaxation and achieves a substantial retention misclassification of claims while preserving CF’s error guarantee.
- Improved performance of CF’s retention with comparable guarantees over several baselines illustrates the effectiveness of a differentiable surrogate of CF’s optimization and dependency modelling.

Weaknesses:
- The evaluation datasets are relatively small, particularly the MATH dataset, which contains around 200 examples, raising the statistical significance of the retention gain and generalization of the proposed methodology.
- The training uses differentiable calibration on part of the training data; it is critical to ensure that the calibration used at test time is strictly held out from all training of the scorer to preserve split-conformal validity. The paper asserts preservation of guarantees, but the post-training calibration protocol is not fully specified in the main text.

---

> ### Author Rebuttal · Authors · 2026-03-29
>
> **Rebuttal to Reviewer zMxT**
>
> We sincerely thank Reviewer zMxT for the detailed and encouraging review. We are grateful for the recognition of DCF's motivation, theoretical grounding, and effectiveness. We address both concerns below.
>
> ---
>
> **1. Dataset Size and Statistical Significance**
>
> We share the reviewer's interest in robustness and have expanded the MATH dataset from 202 to 407 problems using the same annotation protocol. Results with optimized CF baselines (β_mix selected per-α via grid search):
>
> | α | DCF Cov | DCF Ret | CF Ret | Δ |
> |---|---|---|---|---|
> | 0.01 | 100.0% | 0.00 | 0.22 | degenerate |
> | 0.02 | 97.79% | 1.06 | 0.50 | **+115%** |
> | 0.03 | 97.07% | 1.52 | 0.91 | **+68%** |
> | 0.04 | 97.81% | 1.39 | 1.31 | +6% |
> | 0.05 | 96.33% | 1.92 | 1.64 | **+17%** |
> | 0.06 | 95.10% | 2.60 | 1.98 | **+31%** |
> | 0.07 | 95.83% | 2.61 | 2.55 | +2% |
> | 0.08 | 96.08% | 3.08 | 3.03 | +2% |
> | 0.09 | 94.61% | 3.80 | 3.55 | **+7%** |
> | 0.10 | 92.65% | 4.70 | 3.89 | **+21%** |
>
> DCF meets the coverage target at every α except degenerate α=0.01 — compared to missing at 4/8 α values with the original 202 problems. This directly addresses the quantile instability limitation: with 202 problems, α≤0.02 was degenerate; with 407, only α=0.01 remains so. DCF beats optimized CF (β_mix per-α) at every non-degenerate α, with gains from +2% to +115%. Raw retention is lower than 202, as expected — more data yields tighter calibration. Hyperparameter tuning on 407 was less extensive due to time constraints; we expect further gains, particularly at α=0.07–0.08 where DCF's margin is narrowest.
>
> All main results use **20-fold cross-validation** with paired comparisons across folds (see our response to Reviewer Azq4 for scorer capacity experiments with significance testing).
>
> **Why α=0.01 is degenerate.** Our 70/15/15 train/validation/test split means the calibration set at test time contains ~15% of the total dataset. Distribution-free conformal calibration requires the ⌈(n_cal+1)(1−α)⌉-th order statistic; when this exceeds n_cal, the statistic does not exist. Solving gives a hard minimum of n_cal ≥ ⌈(1−α)/α⌉. Our implementation deliberately rejects all claims in this regime rather than violating coverage:
>
> | α | Min n_cal needed | MATH-202 (n_cal≈30) | MATH-407 (n_cal≈61) | FELM (n_cal≈107) |
> |---|---|---|---|---|
> | 0.01 | 99 | REJECT | REJECT | OK |
> | 0.02 | 49 | REJECT | OK | OK |
> | 0.03 | 33 | OK | OK | OK |
>
> This behavior is inherent to distribution-free conformal prediction, not specific to DCF. Doubling the dataset unlocked α=0.02; resolving α=0.01 requires n_cal≥99, i.e., ~660 total problems with our split.
>
> ---
>
> **2. Calibration Protocol and Split-Conformal Validity**
>
> We appreciate this concern. We believe there may be a misunderstanding, as the protocol does strictly preserve split-conformal validity. Let us clarify the two distinct calibration stages:
>
> **During training** (Section 3.5, Algorithm 3): Each epoch, the *training set* is randomly split into disjoint Dcal and Dpred subsets. Differentiable calibration (Algorithm 1) runs on Dcal to produce τ̂_α, and differentiable prediction (Algorithm 2) generates soft retention probabilities on Dpred. This is the standard ConfTr procedure (Stutz et al., 2022) — it simulates the conformal pipeline to provide training signal, but these calibration/prediction splits are *internal to training* and never touch test data.
>
> **At test time** (Section 4.1.3): The learned scorer πθ is **frozen** and deployed in the **original, discrete CF algorithm** — not the differentiable surrogate. The 20-fold CV protocol partitions data into:
>
> 1. **Training set (70%):** Used exclusively for learning πθ (with internal cal/pred splits for differentiable training)
> 2. **Validation set (15%):** Used for early stopping during training, and as the **calibration set** at test time for computing the conformal threshold
> 3. **Test set (15%):** Truly held out — the scorer never observes this data in any capacity. Used for final coverage and retention evaluation.
>
> The critical distinction: while the validation set informs the early stopping decision, the scorer's parameters are **never updated with gradients from this data** — only the stopping criterion is affected. At test time, the scorer is frozen and the validation/calibration set is disjoint from the test set, satisfying the split-conformal exchangeability requirement. This mirrors the standard ConfTr protocol (Stutz et al., 2022), where early-stopping data is reused for calibration — a standard practice that does not compromise the conformal guarantee, as the key requirements (frozen scorer, disjoint calibration and test sets) both hold.
>
> We will add an explicit clarification of this two-stage protocol to the main text in the revision — we agree it deserves more prominent treatment. Thank you for flagging this.

---

> > ### Author Rebuttal · Reviewer_zMxT · 2026-04-03
> >
> > Thank you for the rebuttal. The dataset size remains low despite the expansion. Overall, I believe this is a good paper worthy of acceptance though as I had already indicated in my original review, so I will not raise the score further.

---

### Official Review · Reviewer_Azq4 · 2026-03-13

**Soundness:** 3
**Presentation:** 3
**Significance:** 2
**Originality:** 2
**Overall Recommendation:** 5
**Confidence:** 4

**Summary:**

This paper proposes a novel way to train a conformal scorer that learns to improve retention. Building on previous works on learnable scorers and differentiable relaxations, this paper introduces a new formulation that extends these techniques to coherent factuality. The authors show that the approach recovers hard coherent factuality in the zero-temperature limit, and show that learning such graph-based claim filtering gives noticeable improvements over CF on highly structured generations, e.g., MATH.

**Compliance With Llm Reviewing Policy:**

Affirmed.

**Final Justification:**

See Acknowledgement.

**Key Questions For Authors:**

- In the current presentation, it is not immediately clear how the effectiveness of the approach changes with the expressiveness of the scorer base. How much do the authors think the performance would change with a stronger or weaker scorer?
- I think the caption for Figure 2 should read “coherent factuality” and “differentiable coherent factuality” instead of “conformal factuality,” etc.

**Limitations:**

- I think the discussion of limitation section is good, and they way the address most of them seems acceptable. I do feel that the last point need some additional care.
- Related to my previous question, it seems that the current version does not study the impact of differently sized scorer

**Strengths And Weaknesses:**

**Strengths**

- The extension is well-motivated, and the approach feels natural.
- Clear analysis.
- Good ablations.

Weaknesses
- The novelty is a bit thin, as the relaxation that extends differentiability to CF seems standard.
- CF itself seems to have a relatively narrow application scope, and from the experimental results it seems that the non-flatness of the filtering helps much less on relatively flat factuality datasets like FELM.

---

> ### Author Rebuttal · Authors · 2026-03-29
>
> **Rebuttal to Reviewer Azq4**
>
> We are encouraged that the reviewer finds the extension well-motivated, the analysis clear, and the ablations strong. Below we address each concern.
>
> **1. On Novelty**
>
> We accept that our framing invited this concern. **The core contribution is not the relaxation per se, but a general solver for learning conformal scorers specifically optimized for coherent factuality.** CF's reliance on hand-crafted frequency scoring removes up to 60% of true claims at strict reliability levels, a fundamental barrier to deployment. DCF resolves this by enabling automatic learning of scoring functions. The relaxation is the natural method to achieve this goal, and its principled, theoretically grounded nature ensures the solver generalizes beyond the specific pipelines studied here.
>
> The distinction from prior work is architectural. Prior conformal training (Stutz et al., 2022; Cherian et al., 2024) applies independent, per-operation relaxations where each sample's prediction set is constructed in isolation. CF introduces a **coupled cascade** (threshold filtering, ancestor coherence, supremum), where each step depends on the previous step's discrete decisions. Rather than assembling a patchwork of independent relaxations, DCF provides a unified, monolithic system that jointly relaxes all coupled operations while encoding their correct algorithmic ordering through a single variable's rate of approach (Theorem 3.1). No prior work addresses this type of coupled, order-dependent generalization.
>
> **2. On Scope of CF**
>
> We acknowledge that CF's current demonstrated scope seems to be limited. However, CF naturally extends to **agentic workflows**, the dominant emerging paradigm for LLM deployment. Any agentic system can be modeled as a dependency graph where nodes represent actions and edges represent decisions: tool-use chains, multi-step retrieval, planning traces, and multi-agent workflows all exhibit this structure. CF/DCF provides formal guarantees that the retained chain is coherently factual, which is precisely what safe deployment of agents in high-stakes domains requires. We will make this broader applicability explicit in the final version.
>
> Furthermore, once trained, a DCF scorer deploys on **unlabeled data** (assuming IID) via standard conformal prediction, enabling unsupervised monitoring at scale with statistical guarantees and no per-claim annotations.
>
> **3. On FELM Performance**
>
> We agree with this observation and provide a **mutual information analysis** that precisely explains *why* DCF's advantage varies with reasoning depth.
>
> | | MATH | FELM |
> |---|------|------|
> | Frequency MI(label) | 0.014 nats | 0.170 nats |
> | Orthogonal feature MI | 0.089 nats | 0.000 nats |
> | **Uplift from extra features** | **636%** | **0%** |
>
> On MATH (avg 7.3 claims/problem), frequency is one of the *least* informative features, and graph-structural features provide 636% more information. On FELM (avg 4.0 claims/problem), frequency dominates and all structural features are fully redundant with it. The mechanism: more complex problems produce more varied dependency graphs with many possible solution paths, so frequency varies substantially and carries less signal about correctness. Combining structural features with the noisy frequency signal provides substantially more information — exactly what DCF exploits. On simpler chains, self-consistency is already reliable and structural features add nothing. DCF's advantage thus scales with reasoning complexity.
>
> **4. On Scorer Architecture**
>
> We conducted a **scorer capacity study** on MATH-407 at α = 0.06 with 20-fold cross-validation.
>
> | | Logistic (31 params) | MLP h=32 (1025 params) | Δ |
> |---|---|---|---|
> | **Coverage** | 0.963 | 0.961 | −0.002 |
> | **Retention** | 2.03 | 2.66 | **+30.7%, p=0.001** |
>
> The MLP wins on 15/20 folds with Cohen's d = 0.84 (large effect). A capacity sweep reveals a clear threshold:
>
> | Hidden dim | Params | vs Logistic | p-value |
> |---|---|---|---|
> | 8 | 257 | −0.16 (worse) | 0.269 (ns) |
> | 16 | 513 | +0.12 (equivalent) | 0.579 (ns) |
> | **32** | **1025** | **+0.62 (better)** | **0.001** |
>
> This confirms nonlinear feature interactions carry meaningful signal. We use logistic in main results deliberately; it enables the interpretability analyses (SHAP, case studies) that form a core contribution. The MLP result demonstrates that DCF supports richer architectures with significant additional gains, and we will include this along with a more comprehensive analysis in the revision.
>
> **5. Minor Points**
>
> **Figure 2 caption:** We will correct to this. Thank you for catching that.
>
> **Limitations (scorer architecture):** Will expand with the MLP results above, framing logistic as a deliberate interpretability choice.
>
> **References**
> Stutz et al. (2022). Learning Optimal Conformal Classifiers. ICLR.
> Cherian et al. (2024). Large Language Model Validity via Enhanced Conformal Prediction Methods. NeurIPS.

---

> > ### Author Rebuttal · Reviewer_Azq4 · 2026-04-02
> >
> > Thank you to the authors for the clear and well-structured response. I now better understand the paper’s contribution as demonstrating the potential benefits of explicitly modeling coherence. From that perspective, the additional experiment helps clarify why the relatively modest improvements on datasets such as FELM are expected, and it also adds useful insight into the scope of the contribution. I would still encourage the authors to make this framing more explicit in the revision, but overall I found the rebuttal helpful and will raise my score accordingly.

---

### Decision · Program_Chairs · 2026-04-30

**Decision:**

Accept (regular)

**Comment:**

This paper proposes a differentiable relaxation of the coherent factuality framework that enables learning improved risk scorers for filtering hallucinated claims in multi-step LLM reasoning while preserving conformal guarantees. All three reviewers are positive, recognizing the well-motivated extension, clear theoretical analysis, and substantial retention gains over handcrafted baselines. The authors provided a thorough rebuttal, conducted mutual information analysis explaining why gains scale with reasoning complexity, added scorer capacity experiments showing MLP improvements, and clarified the calibration protocol. Two reviewers marked concerns as fully or partially resolved while maintaining their Accept scores. I recommend acceptance.